# *Keep Garfagnina alive*. An integrated study on patterns of homozygosity, genomic inbreeding, admixture and breed traceability of the Italian Garfagnina goat breed

**Christos Dadousis**[1]*, **Francesca Cecchi**[2], **Michela Ablondi**[3], **Maria Chiara Fabbri**[1], **Alessandra Stella**[4], **Riccardo Bozzi**[1]

**1** Dipartimento di Scienze e Tecnologie Agrarie, Alimentari, Ambientali e Forestali, Università di Firenze, Firenze, Italy, **2** Dipartimento di Scienze Veterinarie, Università di Pisa, Pisa, Italy, **3** Dipartimento di Scienze Medico-Veterinarie, Università di Parma, Parma, Italy, **4** Istituto di Biologia e Biotecnologia Agraria, Consiglio Nazionale delle Ricerche, Milano, Italy

* christos.dadousis@unipr.it

**Data Availability Statement:** The genotype data of Garfagnina goats used in the present study are

## Abstract

The objective of this study was to investigate the genetic diversity of the Garfagnina (GRF) goat, a breed that currently risks extinction. For this purpose, 48 goats were genotyped with the Illumina CaprineSNP50 BeadChip and analyzed together with 214 goats belonging to 9 other Italian breeds (~25 goats/breed), whose genotypes were available from the AdaptMap project [Argentata (ARG), Bionda dell'Adamello (BIO), Ciociara Grigia (CCG), Di Teramo (DIT), Garganica (GAR), Girgentana (GGT), Orobica (ORO), Valdostana (VAL) and Valpassiria (VSS)]. Comparative analyses were conducted on i) runs of homozygosity (ROH), ii) admixture ancestries and iii) the accuracy of breed traceability via discriminant analysis on principal components (DAPC) based on cross-validation. ROH analyses was used to assess the genetic diversity of GRF, while admixture and DAPC to evaluate its relationship to the other breeds. For GRF, common ROH (more than 45% in GRF samples) was detected on CHR 12 at, roughly 50.25–50.94Mbp (ARS1 assembly), which spans the *CENPJ* (centromere protein) and *IL17D* (interleukin 17D) genes. The same area of common ROH was also present in DIT, while a broader region (~49.25–51.94Mbp) was shared among the ARG, CCG, and GGT. Admixture analysis revealed a small region of common ancestry from GRF shared by BIO, VSS, ARG and CCG breeds. The DAPC model yielded 100% assignment success for GRF. Overall, our results support the identification of GRF as a distinct native Italian goat breed. This work can contribute to planning conservation programmes to save GRF from extinction and will improve the understanding of the socio-agro-economic factors related with the farming of GRF.

## Introduction

Local breeds, which usually consist of a small number of animals, are increasingly recognized by E.U. action plans as a key feature of unique rural landscapes and agroecosystems. For example, local breeds i) are rustic and adapted to their local environment, ii) represent a significant

available from the Dryad Repository (doi:10.5061/dryad.jwstqjq73).

**Funding:** This work was supported by grants of the University of Pisa (PRA 2016) and grants of the University of Firenze (RICCARDOBOZZIRICATEN19).

**Competing interests:** The authors have declared that no competing interests exist.

economic resource and have been used for many years in manufacture of niche products, especially in mountainous regions, iii) represent an important and usually unique genetic resource that could be essential to address the future changes in climate or disease outbreaks [1], and iv) play an important role for the preservation of human cultural inheritance. For small ruminants in particular, which can adapt to marginal and difficult production environments, the provision of ecosystem services is of major importance. In mountainous regions of the Mediterranean basin, grazing can be used as a measure of protection against avalanches in winter and fire outbreaks during the summer period. Controlled grazing is a cost-effective, non-polluting, nontoxic, nearly carbon-neutral and effective technique to prevent fire propagation. In this context, goat grazing has been proposed as an eco-friendly solution for wildfire prevention [2]. Moreover, climate change has been considered as an additional challenge to the sustainability of livestock systems (e.g., health and productivity) and local breeds may help overcome this challenge by their ability to adapt to heterogeneity in the regions they are reared. Despite these favourable traits, the relatively low productivity of local unimproved breeds often contributes to low farmer's income and endangers their existence.

The widespread presence and adaptation of goats (*Capra hircus*) to a variety of agro-ecological conditions worldwide is well documented [3]. The demographic history of the domesticated goat relates closely to that of human civilization. Sheep, cattle, pigs and goats were among the earliest domesticated ungulates [4, 5]. Based on the Domestic Animal Diversity Information System (DAD-IS) data, 21 goat breeds have already gone extinct (18 of which were reared in the regions of Europe and the Caucasus) and 41 are in a critical situation (all of which are from Europe and the Caucasus) [6]. In Italy, 3 goat breeds are extinct and 12 more are considered to be critically endangered.

The Garfagnina (GRF) is an Italian goat breed at risk of extinction. The latest assessment (October 2019) reported 1,468 animals from 29 different farms (Associazione Regionale Allevatori Toscana-ARAT, personal communication). GRF is mostly reared for dairy production and is found in in central Italy, specifically in the hills and mountains of the northwest Tuscan Apennine area. The origins of this population are not clear. However, it is very likely that this breed was a result of crossings between native goats from the Alps and from the Tuscan-Emilian Apennines. Moreover, local breeders report that the population has been reared for generations for its milk and meat production [7]. This breed has also been linked to the production of typical products, such as the "Controneria" meat kid and the "Caprino delle Apuane" cheese. As it has been reported by Martini et al. [7], GRF goats are usually milked by hand.

To support the management and conservation of this breed, and to provide support to the farmers and to the general region where the breed is reared, a few studies have investigated i) various production characteristics [7, 8], ii) its adaptive profile (via physiological, haematological, biochemical and hormonal parameters) [9] and iii) its resistance to diseases [10, 11]. Martini et al. [7] investigated various zootechnical characteristics of GRF breed in comparison to other Italian and foreign goat breeds. Based on their results, the authors suggested the development of a breeding scheme based on pure bred animals. Nevertheless, no whole genome analysis has been conducted yet to investigate the genomic background of GRF and its ancestry. Genomic information can quantify the genetic diversity among breeds and identify common ancestry and thus inform the planning of conservation programmes.

The GoatSNP50 BeadChip (http://www.goatgenome.org; [12]) released in 2013, together with the recent and publicly available results of the goat AdaptMap project [3], offered us the opportunity to investigate the genetic diversity of GRF and to carry out comparative analyses with other Italian native goat breeds.

The objectives of the present study were to investigate the genetic diversity of GRF by using single nucleotide polymorphisms (SNP) data and to assess its genetic relationship to the other

native Italian goat breeds included in the AdaptMap dataset. A unified procedure on admixture, discriminant analysis and runs of homozygosity (ROH) was applied. The latter was used to assess the level of genetic diversity of GRF, while admixture and discriminant analysis to evaluate the relationship of GRF to other breeds. In brief, admixture quantifies proportions of ancestries per individual from pre-defined groups, while discriminant analysis classifies a given sample to one of the groups analyzed. Runs of homozygosity have been widely used in livestock to assess the degree of genomic inbreeding ($F_{ROH}$), i.e. estimates of inbreeding coefficients based on molecular data [13–16]. They consist of long consecutive homozygous DNA segments which are distributed along the genome. Overall, our results suggest a distinct genetic pool of GRF with low levels of genomic inbreeding.

## Materials and methods

### Ethics statement

Blood sampling for GRF goats was conducted by veterinarians and no invasive procedures were applied. Thus, in accordance to the 2010/63/EU guide and the adoption of the Law D.L. 04/03/2014, n.26 by the Italian Government, an ethical approval was not required for our study.

### Genomic data

Five millilitres of blood were collected from each of 48 female GRF goats (between 2 and 9 years old; all registered in the herdbook) from a pool of 269 females in the Garfagnina district (Media Valle del Serchio, Lucca, Central Italy). Animals were genotyped with the Illumina GoatSNP50 BeadChip (Illumina Inc., San Diego, CA) containing 53,347 SNPs [17]. Genomic data of nine Italian autochthonous goat breeds, namely Argentata dell'Etna (ARG), Bionda dell'Adamello (BIO), Ciociara Grigia (CCG), Di Teramo (DIT), Garganica (GAR), Girgentana (GGT), Orobica (ORO), Valdostana (VAL) and Valpassiria (VSS) were downloaded from the online repository (https://datadryad.org/stash/dataset/doi:10.5061/dryad.v8g21pt) of the ADAPTmap project [3, 18]. The breeds were selected based on breed abbreviation in the PLINK sample information file (.fam) downloaded from the repository and breed description (code and country) reported in Table 1 of [18]. The two datasets were merged and quality control was conducted in PLINK v1.9 [19, 20] based on the following criteria: i) only autosomes were kept, ii) call rate per SNP >95% and iii) missing values per sample <10%. After editing, 260 samples and 48,716 SNP were retained (Table 1). The distribution of SNP per chromosome (CHR) is presented in S1 Fig.

**Table 1. Names of breeds, breed code and number of animals analyzed before (pre-QC) and after (post-QC) quality control per breed.**

| Breed name | Breed code | No. pre-QC | No. post-QC |
|---|---|---|---|
| Argentata dell'Etna | ARG | 25 | 24 |
| Bionda dell'Adamello | BIO | 24 | 24 |
| Ciociara Grigia | CCG | 19 | 19 |
| Di Teramo | DIT | 24 | 24 |
| Garganica | GAR | 20 | 20 |
| Girgentana | GGT | 30 | 30 |
| Garfagnina | GRF | 48 | 48 |
| Orobica | ORO | 24 | 23 |
| Valdostana | VAL | 24 | 24 |
| Valpassiria | VSS | 24 | 24 |

## Runs of homozygosity

Analysis of ROH was conducted in R (*v. 3.5.0*) using the package *detectRUNS v. 0.9.5* [21, 22]. The consecutive method [23] that runs under the main function *consecutiveRUNS.run* was adopted. The required parameters were set to: i) minimum number of 15 SNPs/ROH, ii) 1 Mbp minimum length of ROH and iii) allow one heterozygous SNP within an ROH (to account for genotyping errors). In addition, ROH lengths were split into five classes (0–2, 2–4, 4–8, 8–16 and >16 Mbp). For each of the class and breed, descriptive statistics of ROH per breed, per chromosome, per SNP and per length class were estimated. Principal component analysis (PCA) was used to identify (dis)similarities among breeds, relative to the average number of ROH identified per chromosome. PCA was applied via a singular value decomposition (*prcomp* function in R [21]). In addition, genomic inbreeding ($F_{ROH}$) was calculated per breed (defined as the length of ROH over the total autosomal length per goat and summed over all goats per breed). Regions with a high frequency of ROH (≥45%) were detected and genes located within ± 1 Mbp were annotated by using the *Capra hircus* ARS1 (http://www.ensembl.org/index.html) and the variant effect predictor (https://www.ensembl.org/Tools/VEP) Ensembl databases.

## Population stratification and ancestry

PCA and admixture analysis were used to infer the presence of distinct populations based on the SNP data. PCA was conducted on the matrix of genotypes. The proportion of mixed ancestry in the breeds was assessed by the *ADMIXTURE 1.22* software [24, 25]. The number of ancestries (K) to be retained in the admixture analysis (K = 2 to 10) was evaluated via a 10-fold cross-validation (CV). The final selection on the number of ancestries was done by inspecting the CV error.

## Discriminant analysis of principal components

Discriminant analysis was applied to assess the breed traceability of the GRF goats using SNP data. To achieve this, the methodology of discriminant analysis of principal components (DAPC) [26] implemented in the *adegenet* R package [21, 27, 28] was adopted. In brief, DAPC is a 2-step approach: firstly, a PCA on the matrix of the genotypes is conducted and then, a small number of selected PCs (instead of the original SNP genotypes) is used as an input for the linear discriminant analysis (LDA). The selection on the optimal number of PCs to be further used in the LDA is done via cross-validation (CV) where the data is split in training and validation sets. The following criteria were implemented for selection of PCs: i) 10-fold CV with 30 repetitions, ii) a maximum number of 259 PCs were tested, and iii) the number of PCs to be retained was based on number of PCs associated with the highest mean success. Three different scenarios of DAPC were applied as described below:

1. Scenario 1 (supervised learning). The full dataset was analyzed simultaneously. In this scenario, all available data were used for model training and the discriminant functions were extracted based on all animals. This is not, however, a real case scenario, since the discriminant functions were developed utilizing the entire data set. The objective for a practical application is to identify an external individual membership to a group (i.e., external validation). Hence, two more scenarios were developed adopting a CV scheme also for the discriminant function.

2. Scenario 2 (semi-supervised learning). Assessment of correct assignment of GRF goats was done via a semi-supervised CV ($CV_{SS}$). Five GRF goats were sampled representing the testing set of the DAPC analysis. The training set (TRN) was constituted by the remaining of

43 GRF goats plus all the goats from the other breeds. The five GRF samples were classified in one of the 10 breeds presented in the TRN set via the function *predict.dapc*. The procedure was repeated 10 times and results were averaged over the 10 repetitions.

3. Scenario 3 (unsupervised learning). Assignment of GRF goats in a breed but without the presence of any GRF goats in the TRN set (unsupervised CV; $CV_{US}$). This scenario could also be viewed as a method to assess the genomic similarity of the GRF with the rest of the breeds (i.e., type of clustering). The approach was similar to Scenario 2 other than the testing population consisted of the entire GRF set and GRF samples had to be classified in one or more of the other 9 breeds. To increase the number of the tested samples in each round of the CV, 80% of GRF breed was sampled. Moreover, to test for the effect of the size in model training for the assignment of GRF, different proportions of TRN set were sampled (20, 30, . . ., 90%) 10 times each, and results were averaged over 10 repetitions. In other words, the size of the TRN set varied between 42 to 191 goats. All nine breeds were present in each scenario and all GRF goats were used in this scenario.

The terms (semi/un)-supervised should not be confused with the terminology used in machine learning. The introduction of these terms has been used in the manuscript to distinguish among the three approaches that were used in the DAPC analysis, and, although they are, up to a point, analogous with the same terms used in machine learning they are not identical.

## Results

### Runs of homozygosity

Summary results of the detected ROH regions as either total counts or averaged based on the number of samples per breed are presented in Table 2 and Fig 1a, respectively. A relatively high number of ROH was detected for GRF (n = 2,450), which was the third largest among the breeds in the study. The greatest number of ROH among all breeds was for GGT (n = 2,762) followed by ORO (= 2,693), while the smallest was found for ARG (n = 465). For GRF, the number of ROH per *Capra hircus* chromosome (CHI) varied from 35 (CHI25) to 158 (CHI1). The maximum length of ROH per chromosome was found on CHI1 (568,887,711bp) and the minimum on CHI23 (113,323,345bp). In general, the total length of ROH per CHR followed the same pattern of the total ROH number per CHR (Fig 1b).

For all breeds analyzed except CCG and DIT, the number of ROH relative to the length on the genome decreased with an increase in length (Fig 1c and 1d). In DIT samples, ROH were

**Table 2. Total number of runs of homozygosity (ROH) detected per breed.**

| Breed | No. ROH |
|---|---|
| ARG | 465 |
| BIO | 814 |
| CCG | 496 |
| DIT | 813 |
| GAR | 730 |
| GGT | 2,762 |
| GRF | 2,450 |
| ORO | 2,693 |
| VAL | 1,613 |
| VSS | 768 |

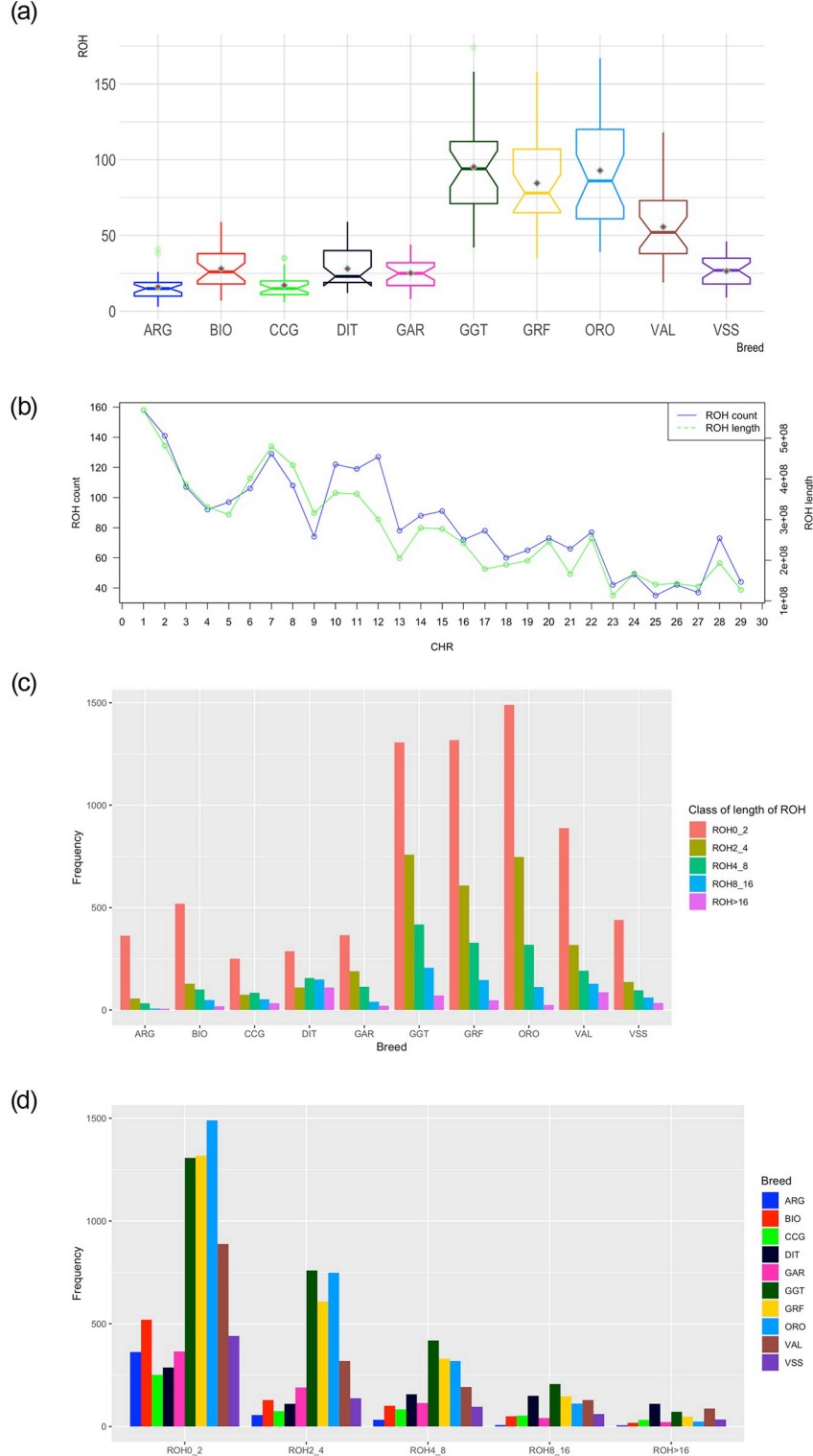

**Fig 1. Summary results of runs of homozygosity (ROH): (a) average number detected per breed, (b) number and length of ROH per chromosome (CHR) in Garfagnina breed, (c) frequency distribution of the number of ROH in the breeds analyzed per length class and (d) frequency distribution of the number of ROH in different length classes per breed.** ARG: Argentata dell'Etna; BIO: Bionda dell'Adamello; CCG: Ciociara Grigia; DIT: Di Teramo; GAR: Garganica; GGT: Girgentana; GRF: Garfagnina; ORO: Orobica; VAL: Valdostana and VSS: Valpassiria; In (a) horizontal bars within each boxplot represent the median, and red rhombus the mean.

more frequent in length classes of 4–8 and 8–16 Mbp compared to 2-4Mbp. The percentage of ROH with a length >16Mbp per breed varied between 0.89% to 13.53% for ORO and DIT, respectively. For small ROH length (<2Mbp) the proportion over the total number detected reached ~78% in ARG, while only 35% of ROH was observed for DIT. The pattern of ROH length class was similar among GRF, GGT and ORO with ~50% of ROH having a length < 2Mbp, ~25% between 2-4Mbp, ~13% between 4-8Mbp, ~5% and ~2% between 8-16Mbp and > 16Mbp, respectively (S1 Table).

For GRF, common ROH (more than 45% in the GRF samples analyzed) were detected on CHI12, between ~34.6–35.3 Mbp (Fig 2, Table 3). In total, 14 SNP were contained in this region. The same area was also present in DIT, while a broader region (~33,9–36.5 Mbp) was shared among ARG, CCG, and GGT. To identify similarities among breeds relative to the number of ROH per chromosome, a PCA was conducted on the average number of ROH identified per chromosome. In addition, a heatmap on the actual number of ROH per chromosome was produced. Both approaches placed GRF closer to ORO and GGT with respect to the rest of the breeds (S2 Fig).

Genomic inbreeding coefficients ($F_{ROH}$) were found to be intermediate for GRF compared to the rest of the breeds analyzed, with a mean value of 0.069 (Fig 3a, S2 Table). The highest values were observed for GGT (0.143) and ORO (0.137). Moreover, the distribution of $F_{ROH}$ calculated per CHR was similar, with some high values (>0.5) observed for CHI7, 9, 16, 22 and 25 (Fig 3b).

## Population stratification and ancestry

As a first step, a PCA was conducted on the complete data to visualize the general structure and relationships among breeds. The first axis distinguished GRF goats from ARG, CCG, DIT, GAR and GGT, while the second axis further separated GRF from the rest of the breeds

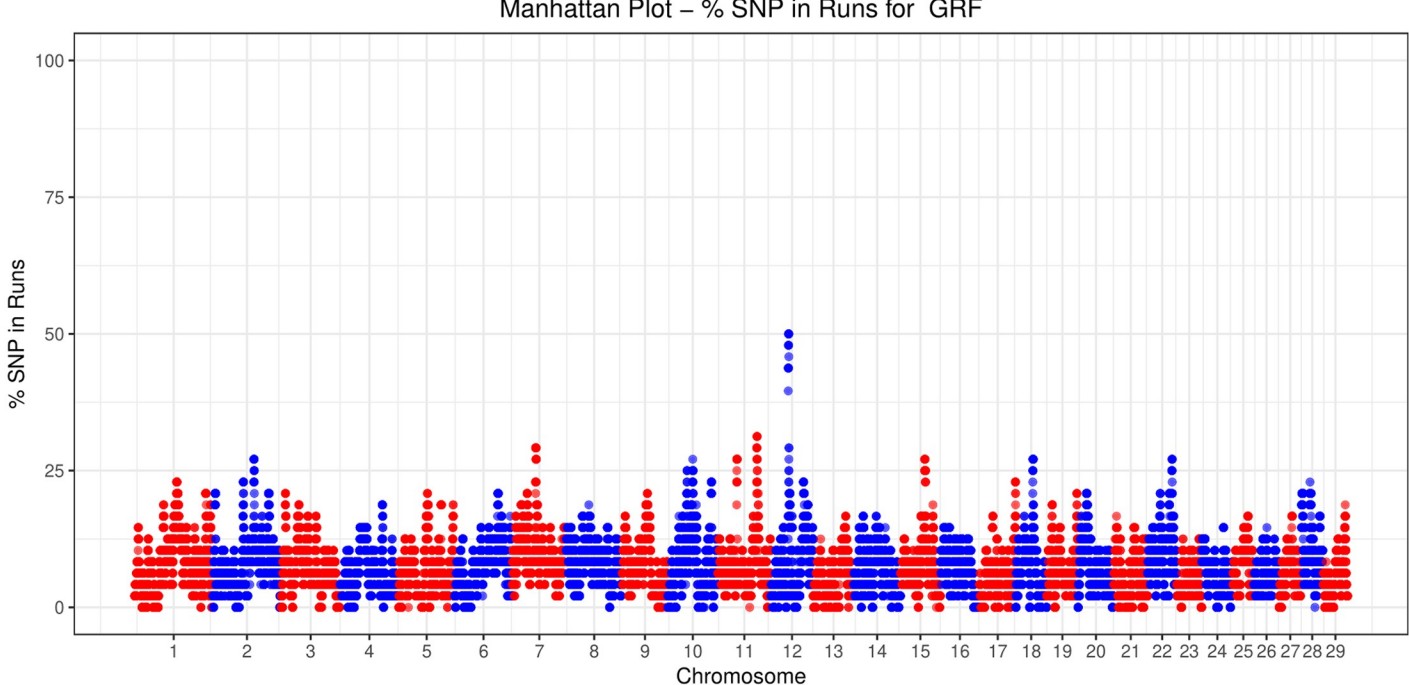

**Fig 2. Number of times (%) each SNP was detected inside a run of homozygosity (ROH) in Garfagnina (GRF) goats.**

**Table 3. Most common (≥45% in each breed) runs of homozygosity (ROH) detected per breed on *Capra hircus* chromosome 12, with the start-end regions and number of SNP per ROH.**

| Breed | Start-SNP | End-SNP | No. SNP | Start-region (bp) | End-region (bp) |
|---|---|---|---|---|---|
| ARG | snp30406-scaffold335-807385/ rs268262652 | snp30399-scaffold335-501928/ rs268262645 | 6 | 34,949,980 | 35,255,437 |
| CCG | snp30421-scaffold335-1558038/ rs268262666 | snp30391-scaffold335-171300/ rs268262637 | 27 | 34,199,327 | 35,586,065 |
| DIT | snp30413-scaffold335-1113038/ rs268262659 | snp30397-scaffold335-418126/ rs268262643 | 14 | 34,644,327 | 35,339,239 |
| GGT | snp30428-scaffold335-1839052/ rs268262673 | snp11142-scaffold140-760668/ rs268243983 | 53 | 33,918,313 | 36,518,132 |
| GGT | snp17454-scaffold1805-39262/ rs268250096 | snp55342-scaffold855-361968/ rs268286936 | 26 | 40,620,364 | 42,257,561 |
| GRF | snp30413-scaffold335-1113038/ rs268262659 | snp30397-scaffold335-418126/ rs268262643 | 14 | 34,644,327 | 35,339,239 |
| VAL | snp50169-scaffold717-4207960/ rs268281880 | snp3193-scaffold1095-4352995/ rs268236233 | 87 | 24,592,901 | 28,744,348 |

(Fig 4). An inspection of all the pairwise comparison between the first 10 axes (PCs) was carried out. In general, GRF was clearly separated from the rest of the breeds. Interestingly, a small GRF subgroup consisting of six goats was observed on the 9th axis (S3 Fig).

An admixture analysis was conducted to complement the PCA results. A varying number of group ancestries was investigated, ranging from K = 2 up to 10. The model with the minimum CV error was the one with eight group ancestries (S4 Fig). In general, the admixture results were in agreement with PCA, depicting the uniqueness of the GRF gene pool (Fig 5a and 5b). At K = 8 (number of ancestries selected after CV) GRF had almost a breed-specific ancestry, with a small percentage of the GRF goats sharing co ancestry with i) BIO and VSS and ii) ARG and CCG and to a small extent with ORO, VAL, DIT and GGT. It should be noted that apart from GRF, group-specific ancestries, at least to a great extent, existed almost for all breeds except ARG, CCG, GAR and VSS.

## Discriminant analysis of principal components

In the first scenario of DAPC, all data were used. The first 40 PCs, explaining ~35.75% of the total variability in the SNP data (S5 Fig), were used in the final DAPC model, resulting in an assignment success rate of 100% for all GRF goats to their breed of origin. The pattern of the genetic diversity based on DAPC is presented in Fig 6, where a clear genetic distance of GRF from the rest of the breeds can be observed.

An external validation scenario, which better reflects a practical application of the discriminant model, was further assessed. In the first analysis (CV$_{SS}$ scenario), GRF had representative animals in the TRN set where the DAPC model was developed. Also, in this case, a 100% correct classification of GRF goats was observed (S3 Table). Interestingly, the classification of GRF was invariant to the number of PCs selected (ranged between 10 to 70) in DAPC (S4 Table). In the second scenario (CV$_{US}$) there were no representative GRF samples in training the model of DAPC. In that case the majority of animals were classified as CCG while few were assigned to DIT in some of the CV replicates (S5 Table). Similar results were obtained with an increased size of TRN set. In the majority of the scenarios, GRF goats were classified either as CCG or DIT, where there were a few cases in which GRF goats were also assigned to VSS, GAR or BIO, but in none of the cases as ORO, GGT or VAL (Fig 7).

## Discussion

There is increasing scientific evidence that genetic diversity of livestock populations is decreasing worldwide [29, 30]. While some domesticated species, such as chicken, sheep and cattle are uniformly distributed across the world, the vast majority of goats (~560 over a total of ~800 millions) are located in Asia, Near and Middle East, while Europe contributes approximately 5% [30], a quarter of which are reared in Greece (estimated data) [31]. On the total European

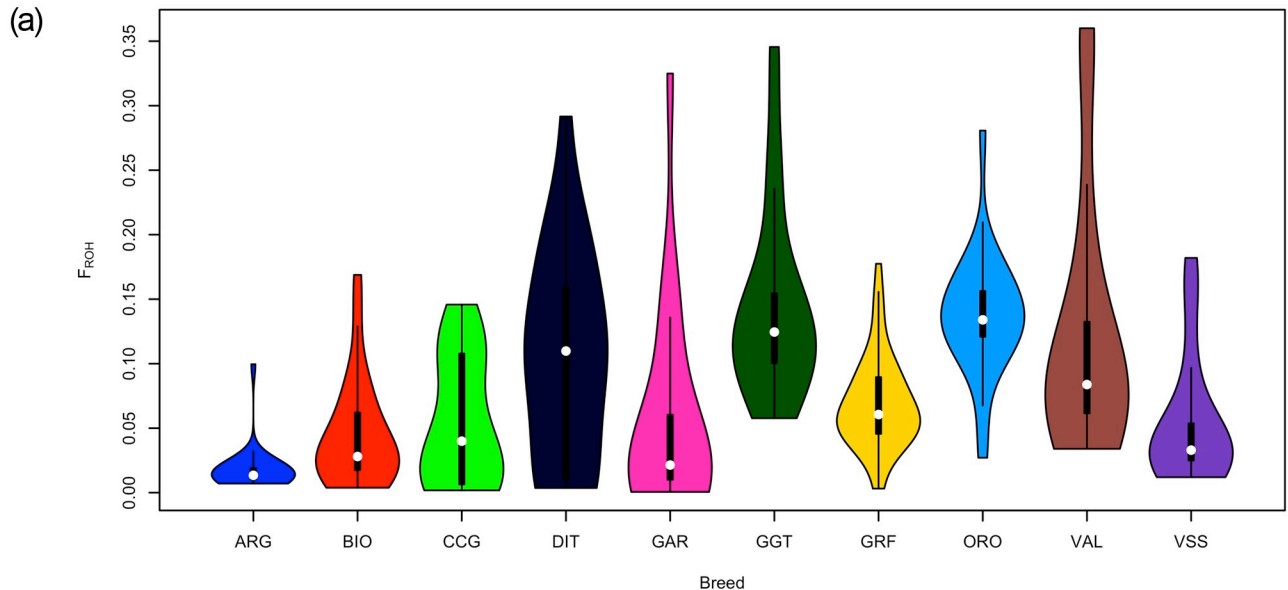

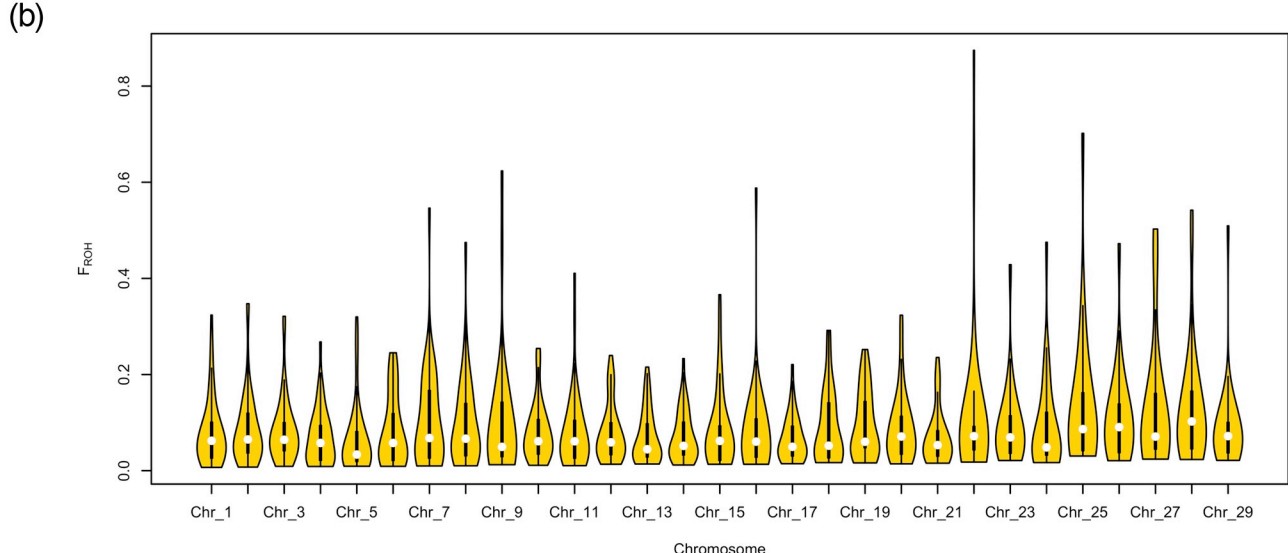

**Fig 3. Summary of the genomic inbreeding coefficients (a) per breed and (b) of the Garfagnina breed per chromosome (Chr).** ARG: Argentata dell'Etna; BIO: Bionda dell'Adamello; CCG: Ciociara Grigia; DIT: Di Teramo; GAR: Garganica; GGT: Girgentana; GRF: Garfagnina; ORO: Orobica; VAL: Valdostana and VSS: Valpassiria.

goat population Italy counts of ~0.5% with ~1 million goats (official data) [31]. The alarming scientific evidence of loss of genetic diversity has led to the development of the Global Plan of Action for Animal Genetic Resources by member countries of the Food and Agriculture Organization (FAO) [32]. Moreover, the concentration of a large proportion of a population in a specific geographical region exposes the breed to the threat of a disease outbreak. This especially holds true for breeds consisting of small populations [30]. Hence, knowledge on the distribution of a breed within a country is critical for policy measures to be taken.

An important question for action measures to be taken to alleviate the problem of a breed extinction, is "*what is a breed*?". As reported by Woolliams *et al.* [33], there is still no clear

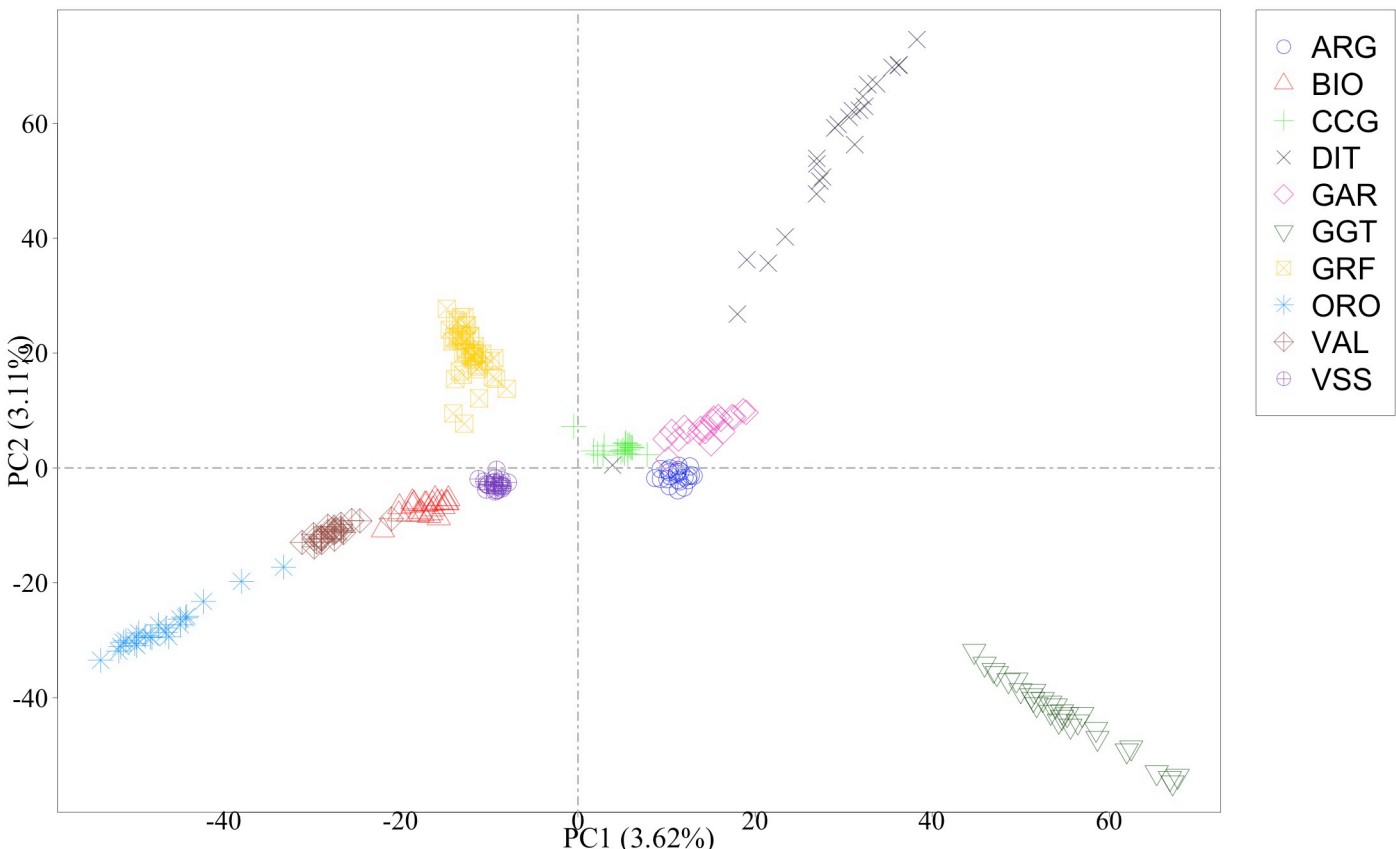

**Fig 4. Scatterplot of the first two principal components[1].** [1]*Singular value decomposition was applied on the matrix of genotypes*. ARG: Argentata dell'Etna; BIO: Bionda dell'Adamello; CCG: Ciociara Grigia; DIT: Di Teramo; GAR: Garganica; GGT: Girgentana; GRF: Garfagnina; ORO: Orobica; VAL: Valdostana and VSS: Valpassiria.

definition of what a breed is. An interesting definition was presented by Hammond (2007) [33]: "*A breed is a breed if enough people say it is*". FAO further outlines the cultural component of a breed. In addition, local livestock populations, generally consisting of small populations and restricted to specific geographical regions, often face difficulties to obtain formal recognition of a breed, since local farmers may not be organized in a breeding association [34]. This difficulty poses further restrictions in policies and funding and, in turn, increases the risk of extinction. Three main pillars have been recently reported for such livestock populations [34]: "*Discover, secure and sustain*". For this purpose, it is essential to conduct studies on quantification of genetic resources of the population in concert with phenotypic and genetic analysis.

GRF is an indigenous goat breed facing a high risk of extinction, with a total number of registered animals less than 1,500. Moreover, its farming is restricted to hills and mountains of the north-western Tuscan Apennine area (central Italy). Given the risk status of the breed, scientists have focused on a better characterization of the GRF population. Various zootechnical parameters of the breed, such as milk total and fatty acid composition, milk coagulation properties and casein genotypes have been previously investigated [7], and authors suggested the development of a purebred breeding scheme. However, no study to date has investigated the genetic diversity of GRF.

(a)

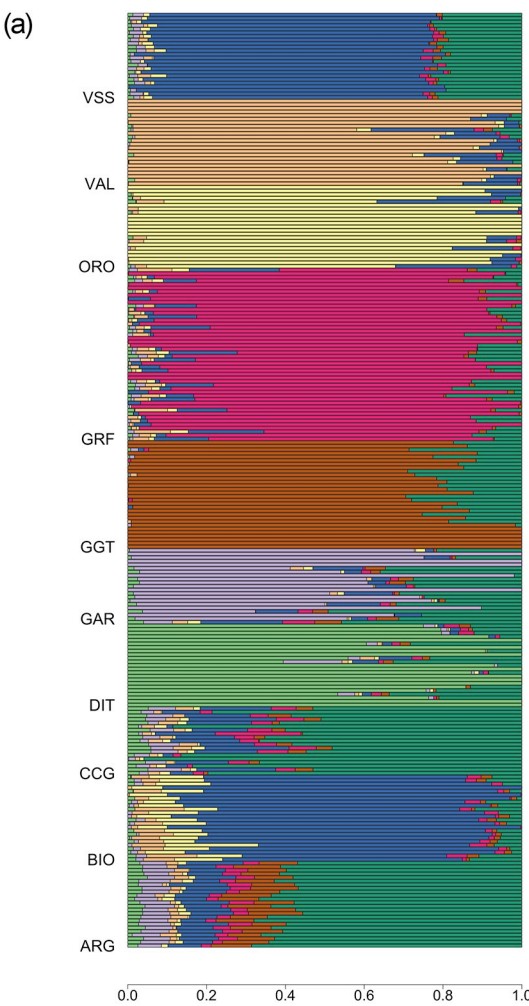

(b)

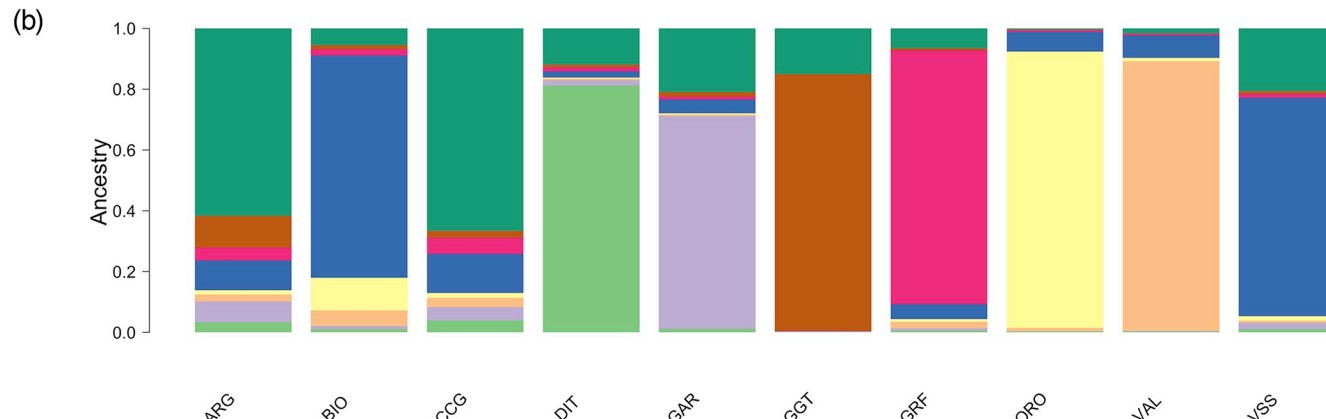

**Fig 5. Admixture analysis with K = 8 coancestry groups (a) per individual and (b) averaged per breed.** ARG: Argentata dell'Etna; BIO: Bionda dell'Adamello; CCG: Ciociara Grigia; DIT: Di Teramo; GAR: Garganica; GGT: Girgentana; GRF: Garfagnina; ORO: Orobica; VAL: Valdostana and VSS: Valpassiria.

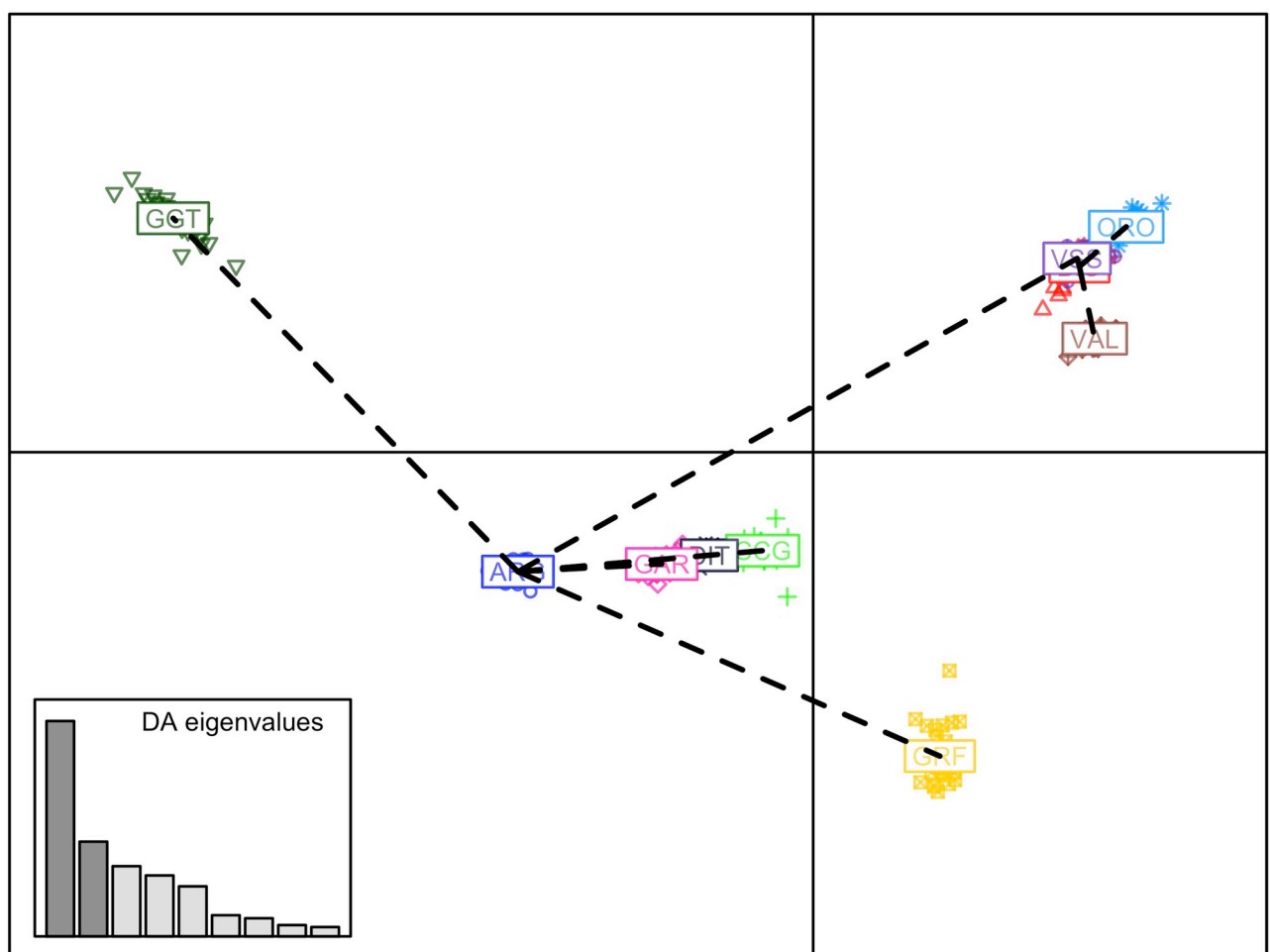

**Fig 6. Scatterplot of the first two discriminant components of the DAPC.** Breeds are presented by different colors and symbols. ARG: Argentata dell'Etna; BIO: Bionda dell'Adamello; CCG: Ciociara Grigia; DIT: Di Teramo; GAR: Garganica; GGT: Girgentana; GRF: Garfagnina; ORO: Orobica; VAL: Valdostana and VSS: Valpassiria.

## Runs of Homozygosity

As it has been highlighted by Bertolini et *al.* [35], crossbred goat populations tend to have smaller total ROH length and number compared to purebred populations. The same pattern has been observed when comparing unselected vs. selected populations undergoing breeding programs. Nevertheless, as it has been pointed out by different colleagues [35, 36] the 50k chip is not considered efficient for the accurate detection of small ROH regions, resulting in underestimation of small ROH hits. Despite this, our analysis was based on purebred goats, and thereby a smaller bias is expected. Results of ROH and $F_{ROH}$ for the 9 Italian breeds of the AdaptMap project were in agreement with results previously reported [35].

The general pattern of ROH (i.e. in terms of total—and by chromosome—number and length of ROH) for GRF was similar to GGT and ORO. Moreover, common ROH were found for GRF (more than 45% in GRF samples analyzed) on CHR 12 at, roughly, between 34.6–35.3 Mbp (Table 3). The same region was also detected in DIT breed, while the broader region (~33,9–36.5 Mbp) was shared among ARG, CCG, and GGT. Further, a search of genes presented in the top ROH region identified for GRF (~50.25–50.94Mbp, updated on the ARS1

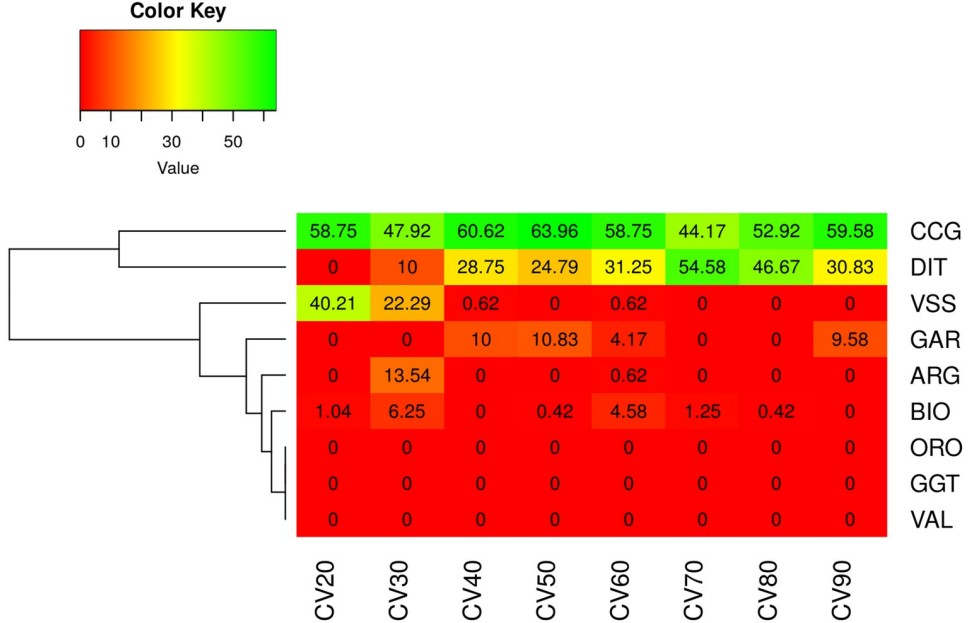

**Fig 7. Percentage of assignment of the GRF goats in the CVUS scenario.** Results were averaged over 10 replicates (CV) in each data subset (from 20 to 90%). CVUS: unsupervised CV, where GRF breed had no representative goats in model training, hence GRF goats had to be classified in one of the other 8 breeds; ARG: Argentata dell'Etna; BIO: Bionda dell'Adamello; CCG: Ciociara Grigia; DIT: Di Teramo; GAR: Garganica; GGT: Girgentana; GRF: Garfagnina; ORO: Orobica; VAL: Valdostana and VSS: Valpassiria.

assembly) and 1Mbp up-downstream (~49.25–51.94Mbp, updated on the ARS1 assembly) was carried out. Interestingly, the region ~49-52Mbp has been previously reported in goats [35, 37]. It is worth noting that within this region lay the genes of the general gap junction protein family *GJA3* (gap junction protein alpha 3; ~50.642–50.644Mbp), *GJB2* (gap junction protein beta 2; ~50.675–50.676Mbp), *GJB6* (gap junction protein beta 6; ~50.694–50.695Mbp). The *GJB2* and *GJB6* are associated with the nervous system, hearing functions and ectodermal processes [38, 39]. Moreover, the *SAP18* (Sin3A associated protein 18; ~51.136–51.141Mbp) that is related to gonad development [40], was also mapped in this region.

The narrow region of the detected top ROH runs for GRF on CHI12 spanned the *CENPJ* (centromere protein J; 50.23–50.27Mb) and the *IL17D* (interleukin 17D; 50.91–50.93 Mb). More precisely, the snp30397-scaffold335-418126 was found to be an intron of the *CENPJ* gene, while the snp30413-scaffold335-1113038 was downstream from *IL17D*. There is a series of studies that have linked the *CENPJ* with primary microcephaly in humans and mice [41–44]. Moreover, *CENPJ* has been found to regulate the neurogenesis and the cilia disassembly in the developing cortex in mice [45]. Also in mouse, disruption of the *CENPJ* can cause the Seckel Syndrome [46]. Possible associations with these physiological functions and phenotypic differences among the breeds studied are not obvious.

Apart from the *CENPJ* and *IL17D* genes, 9 more genes are found within this genomic area, namely *PARP4* (poly(ADP-ribose) polymerase family member 4; ~50.29–50.35Mbp), *MPHOSPH8* (M-phase phosphoprotein 8; ~50.36–50.41Mbp), *PSPC1* (paraspeckle component 1; ~50.44–50.48Mbp), *ZMYM2* (zinc finger MYM-type containing 2; ~50.56–50.63Mbp), as well as the *CRYL1* (crystallin lambda 1; ~50.77–50.83Mbp) and the *IFT88* (intraflagellar transport 88; ~ 50.84–50.91Mbp).

Downstream this region, in 1Mbp expansion, the genes *ATP12A* (ATPase H+/K+ transporting non-gastric alpha2 subunit; ~50.08–50.11Mbp) and *RNF17* (ring finger protein 17;

~50.11–50.23Mbp) were located. Moreover, upstream the region there were also mapped the *EEF1AKMT1* (EEF1A lysine methyltransferase 1; ~50.94–50.95Mbp), *LATS2* (large tumor suppressor kinase 2; ~51.07–51.09Mbp), *ZDHHC20* (zinc finger DHHC-type containing 20; ~51.16–51.22Mbp), *MICU2* (mitochondrial calcium uptake 2; ~51.23–51.28Mbp), and the *FGF9* (fibroblast growth factor 9; ~51.34–51.37Mbp).

## Breed diversity, ancestry and discrimination

Two approaches, complementary to each other, have been used to infer the GRF relationships with 9 native Italian breeds, namely principal component and admixture analysis. Both analyses confirmed the distinct and unique genetic background of the GRF breed and results were in general agreement with each other. Overall, GRF was placed closer to VSS, BIO and CCG, with a small percentage of ancestries shared with all 9 breeds. Interestingly, both approaches identified a subgroup of 6 goats. In admixture, the six goats showed a unique ancestry (Fig 5a), while this subgroup was identified in PCA on the 9th axis (S3 Fig). Nevertheless, the percentage of variance explained by this axis was <1%.

The DAPC model was used to assess breed traceability based on SNP and was able to classify with 100% success GRF goats to their breed of origin (S3 Table). Moreover, an unsupervised learning was applied, where GRF had no representative samples in TRN set. Results were consistent with the PCA and admixture and assigned the majority of GRF goats to the CCG breed with a small number of goats (varied between 4 to 6) assigned as DIT (S5 Table).

As mentioned in the material and methods section, the primary step of the DAPC analysis is to select the number of PCs to be used in the discriminant model. Hence, a basic question emerged on how robust the DAPC could be considered relative to the number of PCs used. Our analysis showed that, although the assignment success was invariant to the number of PCs in the semi-supervised DAPC analysis (number of PCs varied between 10–70), a pattern was found in the case of the unsupervised model. More precisely, when the DAPC contained 40 PCs some of the goats were classified as DIT. In the rest of the cases, where 60 or 20 PCs were used, all GRF goats were assigned as CCG.

PCA analysis seems to detect common ancestries between GRF goats and Alpine Arc goat breeds whereas the DAPC approach identifies similarities between GRF and the goat breeds from Central Italy. Both hypotheses are consistent with the history of Tuscan goat populations that experienced migratory flows from both northern and central Italy.

Overall, the genomic analysis confirmed the hypothesis that GRF is a result of crosses among goats from the Alpine Arc and Tuscan-Emilian Apennines regions. Nevertheless, based on the genomic information analyzed here, GRF represents a unique genetic pool and was genetically distinguished from 9 different native Italian breeds. This, in turn, resulted in breed traceability with a 100% success rate after CV. To sum up, our analysis complements previous work on various zootechnical and adaptive characteristics [7–8] of GRF and provides with a more complete description of the breed.

## Suggestions for monitoring and conservation

One of the strategic priorities of the Global Plan of Action for Animal Genetic Resources of FAO is breed conservation, either *in vivo* or *in vitro* or both [29, 32]. The population status of GRF could be described and summarized in the following points: GRF i) has a distinct genetic pool, ii) exist in a small number of farms and animals, iii) is concentrated in a small geographical region, iv) lacks a formal breeding strategy, v) is reared in semi-extensive and family type farming systems, vi) is not subject to reproductive technologies such as artificial insemination, vii) has low pedigree quality (or almost absent) and viii) lacks of an organized action plan for

monitoring and conservation. These points contribute to the critical status of the breed and highlight the need of conservation policies. For conservation in farms, farmers need to be financially supported by local and international authorities. A direct link between the breed and the production of niche products could add an extra economic value. Moreover, the cultural and the environmental components should be further investigated and quantified in the near future.

In addition, storage of biological material in gene banks offers an extra level of security. As recently been reported by Zomerdijk et al [29], gene banks greatly vary among countries and species and goats come second in order in gene banks after cattle. Even though gene banks, are cost-efficient [47, 48], they face various challenges, such as economic losses, loss of genetic material due to lack of liquid nitrogen and risks from diseases and catastrophic events such as floods and earthquakes [29]. Hence, the storage of biological material in a gene bank should be seen as a complementary rather a substitution to *in vivo* conservation.

Moreover, the idea of the foundation of a GRF breeding association should be considered. Animals are used by farmers if they provide them with profit and profitability is strongly associated to productivity. In this regard, research institutes should provide GRF farmers and breeders with vital scientific support. For example, the development of new tools for breed traceability and recognition might increase farmers' interest. A breeding nucleus, where phenotypic and genetic variability are studied, should be considered. The startup of a breeding scheme, however, should be carefully reflected, since in the Italian legislation, no financial support is provided for livestock conservation if the breed undergoes directional selection, even if the population is at risk of extinction.

Furthermore, genotyping of a large number of animals (males and females) is highly recommended. To reduce costs, genotyping could be targeted, at a first step, only to animals that will contribute to the next generation. In the near future, genomic information could be further utilized for the development of a genomic-based breeding program that would help to select animals early in life and boost the genetic progress. Besides, optimal contribution selection [49, 50], penalization of the number of offspring per male (in the absence of a breeding objective) and a rotational mating scheme, with a controlled buck exchange among farms [51], are some ideas to be considered.

In the three pillars for a successful conservation scheme described in [34] ("*discover, secure and sustain*"), GRF is still under investigation. Various studies have been carried out, while our work presented a first whole-genome scan on the breed's genetic diversity.

## Conclusions

Our genomic analysis suggests a distinct genetic pool of Garfagnina goat breed, with small parts of common ancestry shared with Bionda dell'Adamello, Valpassiria, Argentata dell'Etna and Ciociara Grigia. As a result, GRF can be successfully discriminated by the rest of the breeds analyzed using genomic information with a success rate of 100%. This could further help in a more detailed breed characterization at a genetic level, breed traceability and in controlling the amount of crossbreeding in the future. The above could further support a better monitoring of the breed status. A ROH on CHI12 associated with the *CENPJ* gene should be further investigated in the population. Given the distinct genetic pool, the small number of farms and goats and the restricted farming in a small geographical region, we suggest conservation (*in vivo* and *in vitro*) together with breeding measures to be taken for Garfagnina goat. We hope our work will add value to GRF farming and the local region where the breed is reared.

## Supporting information

**S1 Fig. Number of SNPs per chromosome after quality control. The plot has been produced with the *synbreed R* package [52].**
(DOCX)

**S2 Fig. a) Scatterplot of principal component analysis conducted on the average runs of homozygosity identified per chromosome and breed; b) heatmap on the number of runs of homozygosity identified per chromosome and breed.** ARG: Argentata dell'Etna; BIO: Bionda dell'Adamello; CCG: Ciociara Grigia; DIT: Di Teramo; GAR: Garganica; GGT: Girgentana; GRF: Garfagnina; ORO: Orobica; VAL: Valdostana and VSS: Valpassiria.
(DOCX)

**S3 Fig. Scatterplot of the first 10 principal components[1].** [1]Singular value decomposition was applied on the matrix of genotypes.
(DOCX)

**S4 Fig. Cross-validation results for assessing the number of ancestry groups in the admixture analysis.**
(DOCX)

**S5 Fig. Cross-validation results for the selection of principal components to be retained in the DAPC analysis.**
(DOCX)

**S1 Table. Percentage of the number of runs of homozygosity per length class and breed.** ARG: Argentata dell'Etna; BIO: Bionda dell'Adamello; CCG: Ciociara Grigia; DIT: Di Teramo; GAR: Garganica; GGT: Girgentana; GRF: Garfagnina; ORO: Orobica; VAL: Valdostana and VSS: Valpassiria.
(DOCX)

**S2 Table. Descriptive statistics of the genomic inbreeding coefficients per breed.** ARG: Argentata dell'Etna; BIO: Bionda dell'Adamello; CCG: Ciociara Grigia; DIT: Di Teramo; GAR: Garganica; GGT: Girgentana; GRF: Garfagnina; ORO: Orobica; VAL: Valdostana and VSS: Valpassiria.
(DOCX)

**S3 Table. Assignment results of the Garfagnina goats in the semi-supervised cross-validation ($CV_{SS}$) scenario with 10 repetitions.** ARG: Argentata dell'Etna; BIO: Bionda dell'Adamello; CCG: Ciociara Grigia; DIT: Di Teramo; GAR: Garganica; GGT: Girgentana; GRF: Garfagnina; ORO: Orobica; VAL: Valdostana and VSS: Valpassiria.
(DOCX)

**S4 Table. Number of principal components selected via cross-validation (CV) in the 1st scenario of the DAPC analysis of the Garfganina breed (GRF).** $CV_{SS}$: semi-supervised CV, where some GRF goats are present in model training; $CV_{US}$: unsupervised CV, where GRF breed had no representative goats in model training.
(DOCX)

**S5 Table. Assignment results of the Garfagnina goats in the external cross-validation ($CV_{US}$) scenario with 10 repetitions.** ARG: Argentata dell'Etna; BIO: Bionda dell'Adamello; CCG: Ciociara Grigia; DIT: Di Teramo; GAR: Garganica; GGT: Girgentana; GRF: Garfagnina; ORO: Orobica; VAL: Valdostana and VSS: Valpassiria.
(DOCX)

## Acknowledgments

We thank the two anonymous reviewers for their constructive reviewing. We would also like to thank Dr. Paul Boettcher (FAO, Department of Animal Production and Health Division, Rome, Italy) and Dr. Anna Eleonora Karagianni (The Roslin Institute and the Royal (Dick) School of Veterinary Studies, University of Edinburgh, UK) for reading the manuscript and providing with fruitful comments that improved the readability and quality of this work.

## Author Contributions

**Conceptualization:** Christos Dadousis, Francesca Cecchi, Riccardo Bozzi.

**Data curation:** Christos Dadousis.

**Formal analysis:** Christos Dadousis.

**Methodology:** Christos Dadousis, Michela Ablondi, Alessandra Stella, Riccardo Bozzi.

**Resources:** Francesca Cecchi, Riccardo Bozzi.

**Supervision:** Riccardo Bozzi.

**Visualization:** Christos Dadousis.

**Writing – original draft:** Christos Dadousis.

**Writing – review & editing:** Francesca Cecchi, Michela Ablondi, Maria Chiara Fabbri, Alessandra Stella, Riccardo Bozzi.

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
