## [Decision Letter · Decision Letter 0]

28 May 2020

PONE-D-20-10205

Keep Garfagnina alive. An integrated study on patterns of homozygosity, genomic inbreeding, admixture and breed traceability of the Italian Garfagnina goat breed.

PLOS ONE

Dear Dr. DADOUSIS,

Thank you for submitting your manuscript to PLOS ONE. After careful consideration, we feel that it has merit but does not fully meet PLOS ONE’s publication criteria as it currently stands. Therefore, we invite you to submit a revised version of the manuscript that addresses the points raised during the review process.

There is a number very valid points the reviewers mention that should be implemented in the revised MS to improve readability and clarity. Please pay attention to the discussion to avoid repetitions or results that are mentioned earlier. These points focus on the presentation of the study rather than the data or analysis and should be easy to implement.

We look forward to receiving your revised manuscript.

Kind regards,

Axel Janke

Academic Editor

PLOS ONE

Journal Requirements:

3. We note that you are reporting an analysis of a microarray, next-generation sequencing, or deep sequencing data set. PLOS requires that authors comply with field-specific standards for preparation, recording, and deposition of data in repositories appropriate to their field. Please upload these data to a stable, public repository (such as ArrayExpress, Gene Expression Omnibus (GEO), DNA Data Bank of Japan (DDBJ), NCBI GenBank, NCBI Sequence Read Archive, or EMBL Nucleotide Sequence Database (ENA)). In your revised cover letter, please provide the relevant accession numbers that may be used to access these data. For a full list of recommended repositories, see http://journals.plos.org/plosone/s/data-availability#loc-omics or http://journals.plos.org/plosone/s/data-availability#loc-sequencing

4. In your Methods section, please provide additional details regarding the animals used in your study and ensure you have described the source. For more information regarding PLOS' policy on materials sharing and reporting, see https://journals.plos.org/plosone/s/materials-and-software-sharing#loc-sharing-materials.

5. In your Methods section, please state the volume of the blood samples collected for use in your study.

Reviewers' comments:

Reviewer's Responses to Questions

**Comments to the Author**

1. Is the manuscript technically sound, and do the data support the conclusions?

Reviewer #1: Yes

Reviewer #2: Yes

2. Has the statistical analysis been performed appropriately and rigorously? 

Reviewer #1: Yes

Reviewer #2: Yes

3. Have the authors made all data underlying the findings in their manuscript fully available?

Reviewer #1: Yes

Reviewer #2: Yes

4. Is the manuscript presented in an intelligible fashion and written in standard English?

Reviewer #1: No

Reviewer #2: Yes

5. Review Comments to the Author

Reviewer #1: The paper describes SNP data analyses of an Italian goat breed (Garfagnina). More particularly, the authors have carried out ROH, admixture, PCA and DAPC analyses. In my opinion, data analysis has been sufficiently rigorous, and I especially like the part about DAPC. DAPC is a quite complicated method, and the authors went to lengths to avoid drawing incorrect conclusions from inappropiate settings.

In contrast, in my opinion the presentation of the results is currently not sufficient to justify publication. Points which should be addressed are:

- The text contains many syntax errors. I listed a few of these errors below, but this list represents a random subselection only. One example is consistent misuse of the word 'the'. I suggest all authors to proofread the text prior to resubmission.

- The figures are not ready to publish and should be polished prior to publication. For example, the admixture plot is enormous, the pca/dapc plots are very hard to read (soft colours), and the legends should be inflated to make them readable.

- Number of figures should be reduced, or alternatively figures should be compiled into multi tile figures. For example: the first 4 figures are all on ROHs. Why not summarize the results in one figure?

- Last but not least: for me the overall question, and how the performed analyses address this question, is not clear and should be clarified.

Here a list of minor comments:

Title:

What is ‘genomic inbreeding’?

Abstract:

What do you mean with: ‘genomic background’?

Line 21-22:

I don’t understand the opening sentence: ‘local breeds are recognized as a rule of rural land protection’?

Line 32:

Why not just: ‘Grazing is a cost-effective, nonpolluting…’

Line 34:

Alternative to what?

Line 37:

Incorrect sentence formulation: ‘provide with an alternative’

Line 41-43:

Incorrect sentence

Line 45:

Incorrect: ‘at critical situation’

Line 79:

Specific you are talking about goats used in this study

Line 95-97:

inconsistent sentence construction

Line 105:

Incorrect to say analysis were conducted in a R package

Line 115:

Surrounding genes only? Not overlapping genes?

Line 127:

What do you mean with ‘traceability’?

Line 204:

Even though you specified the details in the methods sections, it would still be helpful for the reader to see the definition of ROH mentioned in this table or in the caption. Also I wonder whether the absolute number of ROHs is really informative. Would it not be more informative to present percentages of the genome, or percentage of number SNPs which are part of the ROHs?

Line 260:

What do you mean with: ‘complement with’?

Line 321:

Consistent incorrect use of ‘the’. For example: it is incorrect to say: ‘following from the above’

Reviewer #2: This study investigated the genomic background of Garfagnina goats, a local goat breed reared in the hills and mountains of the Northwestern Tuscan Apennine area in Central Italy. Despite its cultural and agroeconomic importance, the Garfagnina breed currently faces the risk of extinction and would certainly benefit from a breeding scheme assisted by genomic information. Thus, the research developed here has its value and the methods used are scientifically sound. However, the manuscript doesn’t live up to the standard expected from a full research article and needs to be improved before it is accepted for publication. I listed below some problems that I think should be addressed.

Minor:

- The text is generally well written, but some grammar mistakes are noticeable throughout the text.

- I don’t see a point in showing PCA plots of PC1 x PC2 (Fig. 7a) and PC1 x PC6 (Fig. 7b) while skipping the other principal components (PC3–PC5), specially when the percentage of explained variance is so small. In my opinion, if you don’t have a specific reason for doing that, it is enough to show only the plot for PC1 x PC2 (or a three-dimensional plot with PC1 x PC2 x PC3) in the main text. You can show plots for additional principal components in the supplementary material, if you wish.

- The same can be said about the admixture plots. I don’t see any benefits in showing only K=4 (Fig. 8a) and K=8 (Fig. 8b) in the manuscript. It makes more sense to show the results for all K as a supplementary figure and show only K=8 in the main text, since it is the value of K supported by the cross-validation.

- I also suggest you use a more contrasting color palette for the figures. Particularly in Fig. 8, it is hard to distinguish the ancestry components for some breeds.

Major:

The major problem I see in the manuscript is that a great portion of the Discussion is not serving to its purpose. The Discussion section should be used to put the results into some context, to draw conclusions, and to show their significance and implications. However, the first two paragraphs of discussion are only repeating information that were already stated in the Introduction and Materials & Methods, just using different words.

Another example of that in the Discussion is its section on runs of homozygosity. That whole portion (lines 330-379) can be reduced to a single table of genes present in frequent RoH regions and moved to the Results section. The Discussion should contain what the finding of these genes suggests in the context of the Garfagnina breed.

Moreover, in the Discussion section on population stratification and ancestry, the authors reiterate methodological aspects of the study already described in the Materials & Methods. This redundancy is irrelevant here. Only in the last paragraph, the authors start to build on what should have been the focus of this section since its beginning.

For the reasons mentioned above, I believe that the Discussion needs a major overhaul.

6. PLOS authors have the option to publish the peer review history of their article (what does this mean?). If published, this will include your full peer review and any attached files.

Reviewer #1: No

Reviewer #2: No

---

## [Author Response · Author response to Decision Letter 0]

14 Jul 2020

Title: Keep Garfagnina alive. An integrated study on patterns of homozygosity, genomic

inbreeding, admixture and breed traceability of the Italian Garfagnina goat breed.

AU: We thank the reviewers for their constructive review. Please find below our responses to the points raised. All our responses are preceded by “AU” (and in blue). Track changes were used in the manuscript. Extra changes, not requested by the reviewers, are highlighted in cyan. We hope to find the new version of the manuscript suitable for publication in PLOS ONE.

Looking forward to hearing from you.

Sincerely,

Dr. Christos Dadousis

Academic Editor

Dear Dr. DADOUSIS,

Thank you for submitting your manuscript to PLOS ONE. After careful consideration, we feel that it has merit but does not fully meet PLOS ONE’s publication criteria as it currently stands. Therefore, we invite you to submit a revised version of the manuscript that addresses the points raised during the review process.

There is a number very valid points the reviewers mention that should be implemented in the revised MS to improve readability and clarity. Please pay attention to the discussion to avoid repetitions or results that are mentioned earlier. These points focus on the presentation of the study rather than the data or analysis and should be easy to implement.

AU: We have tried to address all points raised by the reviewers. The Discussion part has been improved for clarity, following the reviewers’ suggestions. The manuscript has been further checked by two external colleagues, acknowledged in the Acknowledgments (L497-502), to improve readability and correct syntax errors.

Reviewer 1

The paper describes SNP data analyses of an Italian goat breed (Garfagnina). More particularly, the authors have carried out ROH, admixture, PCA and DAPC analyses. In my opinion, data analysis has been sufficiently rigorous, and I especially like the part about DAPC. DAPC is a quite complicated method, and the authors went to lengths to avoid drawing incorrect conclusions from inappropiate settings.

In contrast, in my opinion the presentation of the results is currently not sufficient to justify publication. Points which should be addressed are:

- The text contains many syntax errors. I listed a few of these errors below, but this list represents a random subselection only. One example is consistent misuse of the word 'the'. I suggest all authors to proofread the text prior to resubmission.

AU: We have been throughout the manuscript and corrected syntax errors. Moreover, the manuscript has been checked from two external colleagues who are acknowledged in the Acknowledgments (L497-502).

- The figures are not ready to publish and should be polished prior to publication. For example, the admixture plot is enormous, the pca/dapc plots are very hard to read (soft colours), and the legends should be inflated to make them readable.

AU: Indeed, the Figures in the printed pdf version are not in good quality. However, the Figures we have uploaded in PLOS ONE are of high quality. We assume this is a result during the creation of the final pdf file. We are at the availability of the editor to solve the problem.

- Number of figures should be reduced, or alternatively figures should be compiled into multi tile figures. For example: the first 4 figures are all on ROHs. Why not summarize the results in one figure?

AU: Both recommendations have been followed. The Figures of ROH have been placed together and the total number of Figures has been reduced either by excluding or moving Figures to supplements.

- Last but not least: for me the overall question, and how the performed analyses address this question, is not clear and should be clarified.

AU: Several parts have been added or re-arranged for clarity, mainly in the Discussion section (e.g., L81-87, L316-340). Moreover, we added a new subsection proposing measures to be taken for the conservation of the breed, supported by appropriate scientific literature (L438-480).

Minor comments

Title:

What is ‘genomic inbreeding’?

AU: The term genomic inbreeding refers to estimates of inbreeding coefficients of individuals by genomic data rather than pedigree (L84-85). The term is widely known in animal and plant breeding, hence we would like to keep it. As an example, we provide with a list of papers making use of the term “genomic inbreeding” (Gomez-Raya et al., 2015; Mastrangelo et al., 2016; Schiavo et al., 2020; VanRaden et al., 2011). However, following the reviewer’s suggestions other changes in terminology have been done throughout the manuscript for clarity, such as the “genomic background” in the following comment.

Gomez-Raya, L., Rodríguez, C., Barragán, C., Silió, L., 2015. Genomic inbreeding coefficients based on the distribution of the length of runs of homozygosity in a closed line of Iberian pigs. Genet. Sel. Evol. 47, 81. https://doi.org/10.1186/s12711-015-0153-1

Mastrangelo, S., Tolone, M., Gerlando, R.D., Fontanesi, L., Sardina, M.T., Portolano, B., 2016. Genomic inbreeding estimation in small populations: evaluation of runs of homozygosity in three local dairy cattle breeds. animal 10, 746–754. https://doi.org/10.1017/S1751731115002943

Schiavo, G., Bovo, S., Bertolini, F., Tinarelli, S., Dall’Olio, S., Nanni Costa, L., Gallo, M., Fontanesi, L., 2020. Comparative evaluation of genomic inbreeding parameters in seven commercial and autochthonous pig breeds. Anim. Int. J. Anim. Biosci. 14, 910–920. https://doi.org/10.1017/S175173111900332X

VanRaden, P.M., Olson, K.M., Wiggans, G.R., Cole, J.B., Tooker, M.E., 2011. Genomic inbreeding and relationships among Holsteins, Jerseys, and Brown Swiss. J. Dairy Sci. 94, 5673–5682. https://doi.org/10.3168/jds.2011-4500

Abstract:

What do you mean with: ‘genomic background’?

AU: This term is also familiar to the animal/plant breeding community. However, we agree with the reviewer that the term is too general, although the same stands for other terms, such as the very definition of a “breed” (see L329-333 in the manuscript). Hence, we tried to avoid the usage of this term throughout the manuscript, which was mainly replaced by the term “genetic diversity”.

Line 21-22:

I don’t understand the opening sentence: ‘local breeds are recognized as a rule of rural land protection’?

AU: This part has been rephrased (L24-25). Indeed, E.U. and F.A.O. have increased interest in the conservation of local breeds. Apart from the risk of extinction, due to small numbers in many local populations, local breeds have been recognized as a measure of rural land protection and sustainability. The reasons are that local breeds do not only relate to human culture but also they fit better to the local environment. For example, the greater grazing potential of local small ruminants compared to their cosmopolitan competitors, acts as a measure of protection against avalanches in winter and fire outbreaks during the summer period. In addition, it is very common that local breeds are linked to niche and PDO products with geographic origin. These are described in the first paragraph of the Introduction (L30-42).

Line 32:

Why not just: ‘Grazing is a cost-effective, nonpolluting…’

AU: Changed as suggested (L34).

Line 34:

Alternative to what?

AU: The sentence has been rephrased (L36).

Line 37:

Incorrect sentence formulation: ‘provide with an alternative’

AU: The sentence has been rephrased (L36).

Line 41-43:

Incorrect sentence

AU: The sentence has been rephrased (L44-46).

Line 45:

Incorrect: ‘at critical situation’

AU: Changed to “in a critical situation” (L48-49).

Line 79:

Specific you are talking about goats used in this study

AU: Ethics statement relates to the samples of Garfagnina we have sampled.

Line 95-97:

inconsistent sentence construction

AU: AU: The sentence has been rephrased (L106-108).

Line 105:

Incorrect to say analysis were conducted in a R package

AU: The sentence has been rephrased (L118).

Line 115:

Surrounding genes only? Not overlapping genes?

AU: In this part we mean that a search in the whole region (1Mb up/downstream) was followed. Because the word “surrounding” might be misleading to the reader, it has been removed. The sentence has been rephrased to “…were detected and genes located within ± 1 Mbp were annotated …” (L130-131).

Line 127:

What do you mean with ‘traceability’?

AU: Changed to “breed traceability” throughout the manuscript (e.g., L9, 144, 409, 434). 

Line 204:

Even though you specified the details in the methods sections, it would still be helpful for the reader to see the definition of ROH mentioned in this table or in the caption. Also I wonder whether the absolute number of ROHs is really informative. Would it not be more informative to present percentages of the genome, or percentage of number SNPs which are part of the ROHs?

AU: We have introduced a brief explanation of ROH in the Introduction (L83-86). A clear definition of ROH consists from several parts reported in the M&M (L120-122). Hence, we believe that none of these could fit in a Table. Regarding the results on ROH, we have included together with the absolute number, the average number, the length per chromosome, and the distribution per ROH class (Fig 1), and the percentage per length class and breed (S1 Table). In addition, the genomic inbreeding summarizes by definition the length of ROH over the total autosomal length per goat, summed over all goats per breed.

Line 260:

What do you mean with: ‘complement with’?

AU: Admixture and PCA are two approaches very similar and complementary to each other. PCA is widely used to identify structure in the data and to distinguish groups between the samples. In that sense, the objective of PCA is to summarize (dis)similarities over the different groups in the data rather than the individual itself. On the other hand, admixture is focusing on the individuals; it provides with the probabilities of each individual to be clustered in one of the pre-defined group ancestries.

Line 321:

Consistent incorrect use of ‘the’. For example: it is incorrect to say: ‘following from the above’

AU: We would like to thank the reviewer for this observation. We have been throughout the manuscript to correct for syntax errors. Moreover, the manuscript has been checked from two external colleagues who are acknowledged in the Acknowledgments (L497-502).

Reviewer 2

This study investigated the genomic background of Garfagnina goats, a local goat breed reared in the hills and mountains of the Northwestern Tuscan Apennine area in Central Italy. Despite its cultural and agroeconomic importance, the Garfagnina breed currently faces the risk of extinction and would certainly benefit from a breeding scheme assisted by genomic information. Thus, the research developed here has its value and the methods used are scientifically sound. However, the manuscript doesn’t live up to the standard expected from a full research article and needs to be improved before it is accepted for publication. I listed below some problems that I think should be addressed.

The major problem I see in the manuscript is that a great portion of the Discussion is not serving to its purpose. The Discussion section should be used to put the results into some context, to draw conclusions, and to show their significance and implications. However, the first two paragraphs of discussion are only repeating information that were already stated in the Introduction and Materials & Methods, just using different words.

AU: Following your suggestions, we have focused in re-writing the Discussion. A new part has been added at the beginning of the Discussion (L316-340). Moreover, we added a new subsection proposing measures to be taken for the conservation of the breed, supported by appropriate scientific literature (L438-480). Overall, the new adds contribute in clarity and scientific significance of this work.

Another example of that in the Discussion is its section on runs of homozygosity. That whole portion (lines 330-379) can be reduced to a single table of genes present in frequent RoH regions and moved to the Results section. The Discussion should contain what the finding of these genes suggests in the context of the Garfagnina breed.

AU: The first paragraph is discussion on previous studies (L351-359). Moreover, the part between L367-384 describes the gene functionality and reports findings from previous studies. Report on the SNP name and position was included to connect the ROH results with gene annotation. Hence, we do believe that this part has been correctly assigned in the Discussion. We would like to keep this part as is in the manuscript.

Moreover, in the Discussion section on population stratification and ancestry, the authors reiterate methodological aspects of the study already described in the Materials & Methods. This redundancy is irrelevant here. Only in the last paragraph, the authors start to build on what should have been the focus of this section since its beginning.

For the reasons mentioned above, I believe that the Discussion needs a major overhaul.

AU: As mentioned above, a series of changes has been done in the Discussion, improving readability, clarity and the overall quality of the manuscript. We would like to thank the reviewer and hope that we have positively addressed the comment. 

Minor comments

- The text is generally well written, but some grammar mistakes are noticeable throughout the text.

- I don’t see a point in showing PCA plots of PC1 x PC2 (Fig. 7a) and PC1 x PC6 (Fig. 7b) while skipping the other principal components (PC3–PC5), specially when the percentage of explained variance is so small. In my opinion, if you don’t have a specific reason for doing that, it is enough to show only the plot for PC1 x PC2 (or a three-dimensional plot with PC1 x PC2 x PC3) in the main text. You can show plots for additional principal components in the supplementary material, if you wish.

AU: The number of Figures have been reduced based on comments of Reviewer 1. Only the PC1-PC2 plot has been kept in the manuscript. However, we disagree that only the first 2-3 PCs are important. Each eigenvector captures specific variation of the original data. This is the reason why, for example, in many human GWAS studies (multiple linear regression without a relationship matrix) a large number of PCs is kept in the model to correct for population structure. Moreover, we have extended our research into the first 10 PCs. As it is shown, the 9th PC identifies a subgroup of 6 goats in GRF. Not surprisingly, this is in agreement with the admixture analysis (L405-408).

- The same can be said about the admixture plots. I don’t see any benefits in showing only K=4 (Fig. 8a) and K=8 (Fig. 8b) in the manuscript. It makes more sense to show the results for all K as a supplementary figure and show only K=8 in the main text, since it is the value of K supported by the cross-validation.

AU: Only admixture results at K=8 have been kept. 

- I also suggest you use a more contrasting color palette for the figures. Particularly in Fig. 8, it is hard to distinguish the ancestry components for some breeds.

AU: A second plot, averaging the results by breed, has been added (Fig 5b). Indeed, the Figures in the printed pdf version are not in good quality. However, the Figures we have uploaded in PLOS ONE are of high quality. We assume this is a result during the creation of the final pdf file. We are at the availability of the editor to solve the problem.

---

## [Decision Letter · Decision Letter 1]

10 Dec 2020

PONE-D-20-10205R1

Keep Garfagnina alive. An integrated study on patterns of homozygosity, genomic inbreeding, admixture and breed traceability of the Italian Garfagnina goat breed.

PLOS ONE

Dear Dr. DADOUSIS,

Thank you for submitting your manuscript to PLOS ONE. After careful consideration, we feel that it has merit but does not fully meet PLOS ONE’s publication criteria as it currently stands. Therefore, we invite you to submit a revised version of the manuscript that addresses the points raised during the review process.

This revised manuscript is a much improved version, and I thank the authors for their changes. 

Reviewer 1 has new suggestions and comments. The authors should read these over and consider their potential to improve the manuscript, addreessing them as they see fit. Some are simple wording changes/clarifications that can be easily addressed and should be, while others are a matter of personal preference (line 192, line 399) and finally some require some more effort, for example the comment on figure 5. I do not think the authors need to make all changes in order for the paper to be acceptable for publication, and I leave it up to them to decide which ones they want to follow up on. 

However, there are two main points that need to be addressed before acceptance:

1. Data access: PLOSONE and all PLOS journals require authors to make all data necessary to replicate their study’s findings publicly available without restriction at the time of publication. When specific legal or ethical restrictions prohibit public sharing of a data set, authors must indicate how others may obtain access to the data. Although the data is available in ENA, the identifier is not contained anywhere in the actual manuscript text to the best of my ability to find it. Please fix this. 

2. Both reviewers and I are in agreement that the figures are not quite ready for publication yet. First, multipanel figures should be consolidated into single files by the authors. In keeping with reviewer 2's comments, I recommend, but do not demand, that they replot all figures using the sample colour palettes; I notice they've tried to match the ggplot default to the R rainbow palette, but using a single colour scheme through the paper will be easier for readers! I also recommend, but again do not demand, that they use different colour schemes for 1c, such that any given colour means the same throughout the manuscript. 

For the PCA plot in figure 4 I encourage (but once again, do not demand) the authors to increase the size of some of the plotting symbols, and consider their choice of colours, to make sure they're accessible to a broad audience. Likewise, I appreciate the effort that has gone into Supp Fig 3 but given the size of the plot, some of the green groups are hard to distinguish from each other - again, worth considering a slightly more distinctive colour scheme. The rcolorbrewer palettes `set1` or `dark2` (plus black and another colour) might be worth considering. 

3. Finally, while reading the submission I noticed some typographical/wording points:

    line 49: (all of which from Europe and Caucasus) should be (all of which are from Europe and Caucasus) or (all from Europe and Caucasus). Additionally, it should be 'the Caucasus', both here and in line 48.

    line 153: "a maximum number of 300 PCs were tested" The authors have a total of 260 samples, so how are there 300 PCs to test? I believe this is a PCA done on a samples x genotype matrix, so the max number of PCs should be 260? Please clarify this.

    line 211: "For GRF, an excess of frequent ROH (more than 45% in the GRF samples analyzed)" Just clarifying that this means the ROH was seen in over 45% of GRF samples, both here and in other places where the same term is used? In that case, I recommend rewording to "high-frequency" or "common".

    line 336: "This, difficulty, poses" should be "This difficulty poses"

    line 402: "the distinguished and" I think the authors might mean "distinct" instead of distinguished?

    line 442: "GRF i) owes a distinct" I think the authors here mean "has" or "possesses" instead of owes?

We look forward to receiving your revised manuscript.

Kind regards,

Irene Gallego Romero

Academic Editor

PLOS ONE

Reviewers' comments:

Reviewer's Responses to Questions

**Comments to the Author**

1. If the authors have adequately addressed your comments raised in a previous round of review and you feel that this manuscript is now acceptable for publication, you may indicate that here to bypass the “Comments to the Author” section, enter your conflict of interest statement in the “Confidential to Editor” section, and submit your "Accept" recommendation.

Reviewer #1: All comments have been addressed

Reviewer #2: (No Response)

2. Is the manuscript technically sound, and do the data support the conclusions?

Reviewer #1: Yes

Reviewer #2: Yes

3. Has the statistical analysis been performed appropriately and rigorously? 

Reviewer #1: Yes

Reviewer #2: Yes

4. Have the authors made all data underlying the findings in their manuscript fully available?

Reviewer #1: Yes

Reviewer #2: Yes

5. Is the manuscript presented in an intelligible fashion and written in standard English?

Reviewer #1: Yes

Reviewer #2: Yes

6. Review Comments to the Author

Reviewer #1: The authors have addressed all my comments raised in the first revision round, and therefore I have no reason not to accept the paper in the current state. I still think though that the discussion is a bit shallow and that the results could be presented in more concise plots.

Rereading the paper, I had some additional remaining questions and further suggestions for improvement prior to final submission:

Line 4. ‘and analyzed together’

I suggest clarifying here that the data for the 214 goats of the 9 other breeds was not newly generated but taken instead from an online database. This was not initially clear to me.

Line 10: For GRF, an excess of ROH 11 (more than 45% in GRF samples) was detected on CHR 12 at, roughly 50.25-50.94Mbp 12 (ARS1 assembly), which spans the CENPJ (centromere protein) and IL17D (interleukin 13 17D) genes.

Why is it informative to share these specific ROH-details in the abstract?

Line 17: Overall, our results support the identification of GRF as a distinct native Italian goat breed.

Maybe the authors could specify to which analyses ‘overall’ refers to. I suppose particularly admixture analysis and DAPC analyses. The identified ROH in GRF was also present in DIT, and hence does not seem to support identification of GRF as a distinct breed. Perhaps it could be specified in the abstract that DAPC and admixture analyses were conducted to assess the relationship of GRF to other breeds, and that ROH analyses were conducted to assess the genetic diversity of the breed (?).

Line 41:

Check four double spaces throughout document (you could use the replace function in Word to replace all double spaces with single spaces).

Line 87: Overall, our results suggest a distinct genetic pool of GRF

And what do the results suggest about the genetic diversity?

Line 121

Why these parameters? Did you assess the sensitivity of the outcome to the parameter settings? At least a justification is required. In general I am not a big fan of presenting ROH analyses for one parameter setting only, but I am aware many studies do so.

Line 192. Summary results of the detected ROH regions as either total counts or averaged 193 based on the number of samples per breed are presented in Table 2 and Fig 1a, 194 respectively.

This sentence can be deleted. Describe the main results in the text and at the end of the sentences cite the relevant figures.

Line 197.

Why is the distribution per chromosome relevant for the story line?

Line 202-203.

What does the correlation between chromosome length and total ROH length tell about the causal mechanisms behind the ROHs? Simply a chance effect of random distribution of segregating sites across the genome? If so, does it really inform about inbreeding, or is it in fact a measure of genetic diversity?

Line 317-322

Relevant information?

Line 367-397

What is the supposed relation between the ROH and the presence of genomic features? Background selection? I reckon it is not unlikely to detect genomic features in a 2 Mb region. In other words: the presence of genomic features within a ROH is perhaps irrelevant.

Line 399-436

Results section rather than discussion section

Figure 5.

Why not show the outcome for multiple K-values?

Reviewer #2: The manuscript has improved and I'm overall content with the reformulation of the discussion, which was my major concern. However, regarding the quality of the figures, which was also pointed by reviewer 1, the issue is not the resolution of the image but the colors used. We have access to the original image files uploaded by the authors and even in the high resolution images it is hard to distinguish between different breeds in some cases. For instance, I can not easily differentiate two of the three shades of green used in Figure 5 and I'm not color blind. Apart from that, I think the study deserves to be published but the authors must work with the Editor to polish the images before publication.

7. PLOS authors have the option to publish the peer review history of their article (what does this mean?). If published, this will include your full peer review and any attached files.

Reviewer #1: No

Reviewer #2: No

---

## [Author Response · Author response to Decision Letter 1]

21 Dec 2020

AU: We would like to thank again the reviewers and the editor for their constructive review. Please find below our responses to the points raised. All our responses are preceded by “AU” (and in blue). Track changes were used in the manuscript. Extra changes, not requested by the reviewers, are highlighted in cyan. We hope to find the new version of the manuscript suitable for publication in PLOS ONE.

Looking forward to hearing from you.

Sincerely,

Dr. Christos Dadousis

Academic Editor

Dear Dr. DADOUSIS,

Thank you for submitting your manuscript to PLOS ONE. After careful consideration, we feel that it has merit but does not fully meet PLOS ONE’s publication criteria as it currently stands. Therefore, we invite you to submit a revised version of the manuscript that addresses the points raised during the review process.

This revised manuscript is a much improved version, and I thank the authors for their changes. 

Reviewer 1 has new suggestions and comments. The authors should read these over and consider their potential to improve the manuscript, addreessing them as they see fit. Some are simple wording changes/clarifications that can be easily addressed and should be, while others are a matter of personal preference (line 192, line 399) and finally some require some more effort, for example the comment on figure 5. I do not think the authors need to make all changes in order for the paper to be acceptable for publication, and I leave it up to them to decide which ones they want to follow up on. 

AU: We would like to thank the editor for the positive evaluation given to our work.

However, there are two main points that need to be addressed before acceptance:

1. Data access: PLOSONE and all PLOS journals require authors to make all data necessary to replicate their study’s findings publicly available without restriction at the time of publication. When specific legal or ethical restrictions prohibit public sharing of a data set, authors must indicate how others may obtain access to the data. Although the data is available in ENA, the identifier is not contained anywhere in the actual manuscript text to the best of my ability to find it. Please fix this. 

AU: The genotypes of the Garfagnina goats have been deposited on dryad (doi:10.5061/dryad.jwstqjq73). Data availability has been added in the text on L653-655.

2. Both reviewers and I are in agreement that the figures are not quite ready for publication yet. First, multipanel figures should be consolidated into single files by the authors. In keeping with reviewer 2's comments, I recommend, but do not demand, that they replot all figures using the sample colour palettes; I notice they've tried to match the ggplot default to the R rainbow palette, but using a single colour scheme through the paper will be easier for readers! I also recommend, but again do not demand, that they use different colour schemes for 1c, such that any given colour means the same throughout the manuscript. 

AU: We have replotted all figures with different colour schemes while keeping the same colour per breed throughout the manuscript. The multipanel Figures are always uploaded as single files into the system.

For the PCA plot in figure 4 I encourage (but once again, do not demand) the authors to increase the size of some of the plotting symbols, and consider their choice of colours, to make sure they're accessible to a broad audience. Likewise, I appreciate the effort that has gone into Supp Fig 3 but given the size of the plot, some of the green groups are hard to distinguish from each other - again, worth considering a slightly more distinctive colour scheme. The rcolorbrewer palettes `set1` or `dark2` (plus black and another colour) might be worth considering. 

AU: We have replotted Fig 4 increasing the size of the plotting symbols. As above mentioned, we have also changed the colours to better distinguish the different breeds.

3. Finally, while reading the submission I noticed some typographical/wording points:

 line 49: (all of which from Europe and Caucasus) should be (all of which are from Europe and Caucasus) or (all from Europe and Caucasus). Additionally, it should be 'the Caucasus', both here and in line 48.

AU: Changed as suggested (L50-51).

 line 153: "a maximum number of 300 PCs were tested" The authors have a total of 260 samples, so how are there 300 PCs to test? I believe this is a PCA done on a samples x genotype matrix, so the max number of PCs should be 260? Please clarify this.

AU: Thank you very much for this observation, that was a typo and has been corrected (L. 157).

 line 211: "For GRF, an excess of frequent ROH (more than 45% in the GRF samples analyzed)" Just clarifying that this means the ROH was seen in over 45% of GRF samples, both here and in other places where the same term is used? In that case, I recommend rewording to "high-frequency" or "common".

AU: Changed as suggested (L 12, 15, 134, 215, 366).

 line 336: "This, difficulty, poses" should be "This difficulty poses"

AU: Changed as suggested (L 341)

 line 402: "the distinguished and" I think the authors might mean "distinct" instead of distinguished?

AU: Changed as suggested (L 407).

 line 442: "GRF i) owes a distinct" I think the authors here mean "has" or "possesses" instead of owes?

AU: Changed as suggested (L 445)

Reviewer 1

The authors have addressed all my comments raised in the first revision round, and therefore I have no reason not to accept the paper in the current state. I still think though that the discussion is a bit shallow and that the results could be presented in more concise plots.

AU: We would like to thank the reviewer for the general positive evaluation given to our work. 

Rereading the paper, I had some additional remaining questions and further suggestions for improvement prior to final submission:

Line 4. ‘and analyzed together’

I suggest clarifying here that the data for the 214 goats of the 9 other breeds was not newly generated but taken instead from an online database. This was not initially clear to me.

AU: Changed as suggested (L 5-6).

Line 10: For GRF, an excess of ROH 11 (more than 45% in GRF samples) was detected on CHR 12 at, roughly 50.25-50.94Mbp 12 (ARS1 assembly), which spans the CENPJ (centromere protein) and IL17D (interleukin 13 17D) genes.

Why is it informative to share these specific ROH-details in the abstract?

AU: This is also part of the result. Moreover, it is very common to look for genes in ROH analysis that might be related to selection (natural or artificial). The genome assembly is reported because it is strictly linked to the genomic region identification (it is common that between different genome assemblies there might be discrepancies in terms of is re-arrangement of variants/genes within a chromosome or even across chromosomes). Hence, we would like to keep this part in the abstract.

Line 17: Overall, our results support the identification of GRF as a distinct native Italian goat breed.

Maybe the authors could specify to which analyses ‘overall’ refers to. I suppose particularly admixture analysis and DAPC analyses. The identified ROH in GRF was also present in DIT, and hence does not seem to support identification of GRF as a distinct breed. Perhaps it could be specified in the abstract that DAPC and admixture analyses were conducted to assess the relationship of GRF to other breeds, and that ROH analyses were conducted to assess the genetic diversity of the breed (?).

AU: Changed as suggested (L 11-12, 83-84). We would like to keep the term “Overall” on L19.

Line 41:

Check four double spaces throughout document (you could use the replace function in Word to replace all double spaces with single spaces).

AU: Thank you very much for this observation. We made the changes throughout the manuscript.

Line 87: Overall, our results suggest a distinct genetic pool of GRF

And what do the results suggest about the genetic diversity?

AU: The sentence has been extended (L 91) to include also genetic diversity of GRF.

Line 121

Why these parameters? Did you assess the sensitivity of the outcome to the parameter settings? At least a justification is required. In general I am not a big fan of presenting ROH analyses for one parameter setting only, but I am aware many studies do so.

AU: We would like to thank the reviewer for this comment. Indeed, we are aware from literature and own experience of the sensitivity of ROH analyses. In this work we used these parameters that could be considered as “well-established” for this SNP-panel and in line with previous works reporting Froh with the 9 complementary breeds used in our study (L 362-364). We believe that the integrated analysis of admixture, PCA, ROH and DAPC was adequate to provide with a concrete outcome to address our basic question: could we consider GRF as a distinct goat genetic pool in Italy?

ROH analyses provided information of genomic inbreeding and common homozygous segments (with different lengths). Moreover, a comparative analysis was followed, taking into account all breeds. Given that i) genomic inbreeding can be measured with various formulas (e.g, four measures reported in plink, ROH based on sliding windows or consecutive runs and at least one method of constructing GRM) with a range of correlations among these methods and ii) the sensitivity of ROH is not only related to parameters used for defining a ROH but also on SNP QC (MAF, LD and HWE) this type of analysis would be extremely extended and out of the scope of this work. 

Moreover, please note that the same question goes also for DAPC for which we decided to perform an extensive analysis.

Line 192. Summary results of the detected ROH regions as either total counts or averaged 193 based on the number of samples per breed are presented in Table 2 and Fig 1a, 194 respectively.

This sentence can be deleted. Describe the main results in the text and at the end of the sentences cite the relevant figures.

AU: We would like to keep this sentence since it directly provides to the reader important information to follow ROH results.

Line 197.

Why is the distribution per chromosome relevant for the story line?

AU: The distribution of chromosome is relevant because: i) different chromosomes have different SNP densities (S1 Fig) and ii) because a frequent number of ROH on a specific chromosome might be linked with natural or artificial selection. 

Line 202-203.

What does the correlation between chromosome length and total ROH length tell about the causal mechanisms behind the ROHs? Simply a chance effect of random distribution of segregating sites across the genome? If so, does it really inform about inbreeding, or is it in fact a measure of genetic diversity?

AU: Please, see our reply above. Regarding the relationship between inbreeding and genetic diversity, these factors are inter-related and difficult to distinguish, since inbreeding is already a measure of genetic diversity. In general, ROH caused by inbreeding tend to be distributed unevenly over the genome, with a different distribution of numbers and sizes of ROH for each chromosome (Pemberton et al, 2012).

Pemberton, T.J.; Absher, D.; Feldman, M.W.; Myers, R.M.; Rosenberg, N.A.; Li, J.Z. Genomic patterns of homozygosity in worldwide human populations. Am. J. Hum. Genet. 2012, 91, 275–292.

Line 317-322

Relevant information?

AU: We find as relevant information to report the general distribution of goats and would like to keep this part in the manuscript. Moreover, this part is used to open the Discussion section. 

Line 367-397

What is the supposed relation between the ROH and the presence of genomic features? Background selection? I reckon it is not unlikely to detect genomic features in a 2 Mb region. In other words: the presence of genomic features within a ROH is perhaps irrelevant.

AU: ROH that are located in specific genomic regions and shared among several individuals are thought to be potential signs of selection (please see literature below).

Metzger, J.; Karwath, M.; Tonda, R.; Beltran, S.; Águeda, L.; Gut, M.; Gut, I.G.; Distl, O. Runs of homozygosity reveal signatures of positive selection for reproduction traits in breed and non-breed horses. BMC Genom. 2015, 16, 764.

Ablondi, M.; Viklund, Å.; Lindgren, G.; Eriksson, S.; Mikko, S. Signatures of selection in the genome of Swedish warmblood horses selected for sport performance. BMC Genom. 2019, 20,717.

Nolte, W.; Thaller, G.; Kuehn, C. Selection signatures in four German warmblood horse breeds: Tracing breeding history in the modern sport horse. PLoS ONE 2019, 14, 1–25.

Pemberton, T.J.; Absher, D.; Feldman, M.W.; Myers, R.M.; Rosenberg, N.A.; Li, J.Z. Genomic patterns of homozygosity in worldwide human populations. Am. J. Hum. Genet. 2012, 91, 275–292.

Line 399-436

Results section rather than discussion section

AU: We made slight changes in this part, but we would like to keep this subheading, since a comparison of the results obtained from different methods is discussed, information that we find interesting. Regarding DAPC, a discussion on the robustness of the model is provided. This information is not reported in the Results.

Figure 5.

Why not show the outcome for multiple K-values?

AU: That was a request by Rev2 in review round 1:

“The same can be said about the admixture plots. I don’t see any benefits in showing only K=4 (Fig. 8a) and K=8 (Fig. 8b) in the manuscript. It makes more sense to show the results for all K as a supplementary figure and show only K=8 in the main text, since it is the value of K supported by the cross-validation.”

Reviewer 2

The manuscript has improved and I'm overall content with the reformulation of the discussion, which was my major concern. However, regarding the quality of the figures, which was also pointed by reviewer 1, the issue is not the resolution of the image but the colors used. We have access to the original image files uploaded by the authors and even in the high resolution images it is hard to distinguish between different breeds in some cases. For instance, I can not easily differentiate two of the three shades of green used in Figure 5 and I'm not color blind. Apart from that, I think the study deserves to be published but the authors must work with the Editor to polish the images before publication..

AU: We would like to thank the reviewer for the positive evaluation given to our work. We have replotted all figures with different colour schemes while keeping the same colour per breed throughout the manuscript.

---

## [Editor Report · Decision Letter 2]

22 Dec 2020

Keep Garfagnina alive. An integrated study on patterns of homozygosity, genomic inbreeding, admixture and breed traceability of the Italian Garfagnina goat breed.

PONE-D-20-10205R2

Dear Dr. DADOUSIS,

We’re pleased to inform you that your manuscript has been judged scientifically suitable for publication and will be formally accepted for publication once it meets all outstanding technical requirements.

Kind regards,

Irene Gallego Romero

Academic Editor

PLOS ONE
---

## [Editor Report · Acceptance letter]

28 Dec 2020

PONE-D-20-10205R2 

*Keep Garfagnina alive.* An integrated study on patterns of homozygosity, genomic inbreeding, admixture and breed traceability of the Italian Garfagnina goat breed. 

Dear Dr. Dadousis:

I'm pleased to inform you that your manuscript has been deemed suitable for publication in PLOS ONE. Congratulations! Your manuscript is now with our production department. 

Kind regards, 

on behalf of

Dr. Irene Gallego Romero 

Academic Editor

PLOS ONE